# Lysine-Specific Demethylase 1 (LSD1) epigenetically controls osteoblast differentiation

Petri Rummukainen[1,☯], Kati Tarkkonen[1,☯], Amel Dudakovic[2,3], Rana Al-Majidi[1], Vappu Nieminen-Pihala[1], Cristina Valensisi[4], R. David Hawkins[4], Andre J. van Wijnen[2,3,5]*, Riku Kiviranta[1,6]*

1 Institute of Biomedicine, University of Turku, Turku, Finland, 2 Department of Orthopedic Surgery, Mayo Clinic, Rochester, MN, United States of America, 3 Department of Biochemistry & Molecular Biology, Mayo Clinic, Rochester, MN, United States of America, 4 Division of Medical Genetics, Department of Medicine, University of Washington, Seattle, WA, United States of America, 5 Department of Biochemistry, University of Vermont, Burlington, VT, United States of America, 6 Department of Endocrinology, Turku University Hospital, Turku, Finland

☯ These authors contributed equally to this work.
* andre.vanwijnen@uvm.edu (AJW); rikkiv@utu.fi (RK)

**Data Availability Statement:** All sequencing files are available from the GEO database (accession numbers GSE186832, GSE186665).

## Abstract

Epigenetic mechanisms regulate osteogenic lineage differentiation of mesenchymal stromal cells. Histone methylation is controlled by multiple lysine demethylases and is an important step in controlling local chromatin structure and gene expression. Here, we show that the lysine-specific histone demethylase Kdm1A/Lsd1 is abundantly expressed in osteoblasts and that its suppression impairs osteoblast differentiation and bone nodule formation in vitro. Although Lsd1 knockdown did not affect global H3K4 methylation levels, genome-wide ChIP-Seq analysis revealed high levels of Lsd1 at gene promoters and its binding was associated with di- and tri-methylation of histone 3 at lysine 4 (H3K4me2 and H3K4me3). Lsd1 binding sites in osteoblastic cells were enriched for the Runx2 consensus motif suggesting a functional link between the two proteins. Importantly, inhibition of Lsd1 activity decreased osteoblast activity in vivo. In support, mesenchymal-targeted knockdown of Lsd1 led to decreased osteoblast activity and disrupted primary spongiosa ossification and reorganization in vivo. Together, our studies demonstrate that Lsd1 occupies Runx2-binding cites at H3K4me2 and H3K4me3 and its activity is required for proper bone formation.

## Introduction

In undifferentiated pluripotent embryonic stem cells (ESCs), chromatin is in an open configuration as visualized by electron microscopy [1], but chromatin becomes progressively packed and reorganized into heterochromatin upon cell differentiation [2]. Regulation of chromatin status is mediated through DNA methylation and post-translational modifications of histone proteins [3]. The loss of pluripotency and chromatin accessibility is accompanied by increased DNA methylation and deposition of suppressive histone marks [2, 4]. Yet, differentiation

**Funding:** RK: 298625, Academy of Finland, https://www.aka.fi/ RK: 268535, Academy of Finland, https://www.aka.fi/ RK: 139165, Academy of Finland, https://www.aka.fi/ AvW: R01-AR09069, National Institutes of Health, https://www.nih.gov/ The funders had no role in study design, data collection and analysis, decision to publish, or preparation of the manuscript.

**Competing interests:** The authors have declared that no competing interests exist.

requires open state chromatin and gain of active histone marks on the loci for lineage specific genes and their regulatory regions to allow appropriate gene expression and progression to terminally differentiated cell types [2]. Histone methylation, which can occur on multiple lysine and arginine residues in each of the histone proteins, controls both active and inactive chromatin states. For example, tri-methylation of H3K27 or H3K9 maintains repressive closed heterochromatin, whereas methylation of H3K4 is associated with permissive open euchromatin [5]. Methylated histones regulate chromatin structure by providing a code for "reader" proteins, which include chromodomain proteins [6] that have the capacity to recognize defined methylation states to determine functional outcomes [7].

Adult somatic stem cells are distinct from ESCs and may have distinct chromatin states compared to pluripotent and differentiated cells [8]. Osteoblasts originate from skeletal stem cells in vivo [9, 10], but can also emerge from mesenchymal stromal cells (MSCs) that also give rise to chondrocytes, adipocytes and myocytes [11]. The few studies that have examined the epigenetic landscape during differentiation into specific mesenchymal lineages have shown that histone modifications dynamically change during differentiation, whereas there are only modest changes in the methylation of promoter DNA [12–14]. Expression screens for epigenetic regulators in MSCs and osteoblasts have revealed that specific isoforms of both histone H3 and H4 lysine methyl transferases and the corresponding demethylases are robustly expressed during osteoblast differentiation [15–18]. Chromatin state dynamics have been studied in primary murine bone marrow-derived MSCs during osteoblast differentiation [19, 20], and even if the suppressive H3K27me3 mark was found to decrease, only modest changes in the active H3K4me3 mark were observed at gene promoters. This finding is consistent with previous data showing that the deposition of H3K27me3 by methyltransferase Ezh2, the catalytic unit of the polycomb repressive unit 2 (PRC2), regulates MSC commitment to osteogenic and adipogenic lineages [16, 21–27]. Accordingly, downregulation of Ezh2 by pharmacological inhibition, RNA interference or knockout permits transcription of osteoblastic genes, enhances osteoblast differentiation in vitro and bone formation in vivo [25, 26, 28–33] as well as promotes mineralization of dental tissues [34], while not affecting on tendon differentiation [35].

Histone lysine demethylases (KDMs) play a key role in regulating histone methylation (e.g., H3K27me3, H3K9me3 and H3K4me3). KDMs belong to two distinct enzyme families. KDM1A/LSD1 and KDM1B/LSD2 are monoamine oxidases (MAOs), which utilize flavin adenine dinucleotide (FAD) -dependent mechanism to demethylate H3K4 or H3K9 [36]. All other histone lysine demethylases are members of the Jumonji (JmjC)–domain containing 2-oxoglutarate oxygenase family with various subfamilies (KDM2-8), which utilize a Fe(IV)-oxoferryl species to catalyze demethylation reactions of various methylated states of H3K4, H3K9, H3K27 and H3K36 [37]. Several JmjC-demethylases, including nucleolar protein 66 (NO66) [38, 39], Kdm4B/Jmjd2B [40], Kdm6B/Jmjd3 [40, 41], Kdm6A/Utx [22] and Kdm5B/Jarid1B [42], have been shown to regulate osteogenic differentiation.

Within the MAO family, LSD1 has been extensively studied in stem cell biology and development [43, 44], and has also been implicated in the differentiation of a number of cell types including neurons [45, 46], adipocytes [47], chondrocytes [48, 49] and hematopoietic cells [50, 51]. In adipose tissue, LSD1 plays a major role in thermogenesis and in regulating brown adipocyte phenotype [52, 53]. The complex role of LSD1 in cell differentiation has been demonstrated in the pituitary, where LSD1 is involved in transcriptional repression and activation complexes, leading to temporal control of genes regulating organogenesis [54]. Mechanistically, LSD1 demethylates H3K4, leading to transcriptional deactivation of genes in close vicinity [55, 56]. In addition, LSD1 may act on H3K9, causing transcriptional activation when forming a complex with androgen and estrogen receptors [57–59]. Importantly, LSD1 has also been shown to act on non-histone substrates such as p53/TP53 in the DNA damage response

[60] and the DNA methyltransferase DNMT1, through which LSD1 can regulate DNA methylation [23]. Temporal control of both LSD1 and its interaction partners are important for determining the genomic targets and context-specific action of LSD1.

In the present study, we identified Lsd1 among the most highly expressed KDMs in osteoblasts and studied its role in osteoblastogenesis, predominantly binding to promoters of genomic targets with increasing occupancy during differentiation. We found that the pharmacological inhibition of the intrinsic demethylation activity of Lsd1 decreased osteoblast function and trabecular bone volume in vivo, possibly through modulating the chromatin status of Runx2 target sites. Furthermore, the conditional loss of the *Kdm1a* gene encoding for Lsd1 in limb bud mesenchyme in $\mathrm{Lsd1}^{-/-}_{\mathrm{Prrx1}}$ mice led to impaired bone maturation and bone remodeling, as well as cellular disorganization in growth plate.

## Materials and methods

### Cell culture

Mouse MC3T3-E1 cells were purchased from ATCC (MC3T3-E1 Subclone 4, CRL-2593) and maintained in αMEM supplemented with 100 U/ml penicillin-streptomycin, 10% fetal calf serum and 2 mM L-glutamine (maintenance medium). For differentiation experiments, the cells were plated on 6- or 12-well plates and at confluence stimulated by osteogenic medium containing 50 μg/ml ascorbic acid and 10 mM β-glycerophosphate for 7,14 and 21 days. The medium was changed every 2–3 days. Primary calvarial osteoblasts were isolated from newborn mouse calvaria by sequential digestions as described previously [61]. In brief, osteoblasts were derived from newborn mouse calvariae with five consecutive 20 min digestions with 1% collagenase and 2% dispase in αMEM at +37˚C. Fractions 2–5 were pooled. After one passage, cells were plated on 6-well plates in αMEM supplemented with 100 U/ml penicillin-streptomycin and 10% fetal calf serum. Differentiation was induced at confluence by 10 nM dexamethasone, 50 μg/ml ascorbic acid and 10 mM β-glycerophosphate for 7,14 and 21 days. For Lsd1 inhibition studies, MC3T3-E1 cells were treated with selective Lsd1 inhibitors RN-1 and GSK2780854A (GSK-LSD1) or vehicle (DMSO) supplemented medium. Proliferation studies were performed by plating 1000 cells/well in a 96-well plate in maintenance medium and imaging the confluence of the wells with Incucyte S3 analyzer (Essen Bioscience, UK).

### Alkaline phosphatase (ALP) and Von Kossa staining

Cell culture plate wells were fixed for 15 minutes in 10% phosphate buffered formalin. Fixed wells were stained for ALP activity for 45 minutes in RT with freshly filtered ALP staining solution (0,1mg/ml Naphtol AS MX-PO4; 0,6mg/ml Fast Blue RR salt; 0,4% DMF in 0,1M Tris-HCl pH 8,3) and rinsed with water. For von Kossa staining, cell cultures were stained with 2,5% silver nitrate in RT for 30 minutes.

### shRNA silencing

Lentiviral particles containing Sigma TRCI library shRNAs against Lsd1 (TRCN0000071373–71377) and control shRNA (shScrambled, SHC002) were obtained from the Functional Genomics Unit at the University of Helsinki. Mouse MC3T3-E1 cells were transduced with 5 different clones of shLSD1 lentiviruses and shScrambled control vector on 6-well plates with 8μg/ml polybrene overnight. Transduction of cells was confirmed with 5μg/ml puromycin selection for 5 days. The efficiency of Lsd1 mRNA and protein knockdown was confirmed using RT-PCR and Western blotting, respectively.

## Reverse transcriptase quantitative PCR (RT-qPCR) and next generation RNA-sequencing (RNA-seq)

RNA was extracted from cell culture samples with Nucleospin RNA Plus kit (Macherey-Nagel) after homogenization in RNA lysis buffer using Ultra-Turrax T25 homogenizer (Janke&Kunkel, Germany) and eluted to 2x30μl aqua. From bone samples, RNA was extracted using RNeasy Mini kit (Qiagen, USA) after pulverization of snap-frozen bones followed by homogenization in RNA lysis buffer using Ultra-Turraz T25 homogenizer. RNA concentrations were measured with NanoDrop One$^C$ (Thermo Scientific) and the samples were stored at -80˚C. For RT-qPCR, RNA was reverse transcribed to cDNA using the SensiFast cDNA synthesis kit (Bioline, USA) with 500ng RNA per reaction. qPCR reactions were performed using SYBR Green master mix (Thermo Scientific) with the Bio-Rad CFX96 or CFX384 Real-Time PCR Detection System and software. Ct values were normalized to beta-actin (ACTB) levels and ΔΔ-Ct method was used to analyze relative gene expression levels. Primer sequences are shown in S1 Table. RNA-seq was performed on the Illumina2000 platform as 51 bp paired-end reads and processed using a standardized bioinformatic pipeline as previously described in detail [32, 62–64]. Gene expression is expressed in fragments/kilobase pair/million mapped reads (FPKM). RNA-seq data were deposited in the Gene Expression Omnibus of the National Institute for Biotechnology Information (GSE186832).

## ChIP-seq and bioinformatics analysis

MC3T3-E1 cells (10,000 cells/cm2) were plated in 10 cm plates in maintenance medium. The cells were harvested by trypsin and analyzed using a ChIP assay as described previously [21] using Lsd1, H3K4me1, H3K4me2, H3K4me3, H3K27me3 and control IgG antibodies. Antibody information can be found in S1 Table. Sequencing libraries were prepared and massively parallel high throughput DNA sequencing was performed on an Illumina HiSeq2000 system. The alignment, quality assessment, peak calling, and visualization was performed with a standardized bioinformatic analysis pipeline (HiChIP) as described [21]. The mm10 reference genome was used to align 50 base paired-end with a custom script that only retains pairs with one or both ends uniquely mapped. The analysis was guided by the Burrows-Wheeler Aligner, Picard and SAMTools and the SICER package [21]. ChIP-seq data were deposited with accession number GSE186665.

Peak data were read using custom R scripts, and exploratory analysis and peak annotation was done using R package ChIPseeker [65]. Average profiles of ChIP-seq peaks binding to transcription start site (TSS) regions were inspected and visualized, using 3kb upstream and downstream TSS flanking sequences for the mapping. Peaks were annotated to the nearest TSS, with TSS region defined from -3kb to +3kb (default). R package DiffBind [66] was used for generation of the correlation heatmaps. The overlaps of the ChIP-seq peak intervals were inspected using R package ChIPpeakAnno [67] with minimum overlap of 1 bp of the peak intervals required. Further exploratory analysis was performed using DiffBind package. Shortly, consensus peak sets were generated for each of the five ChIP-seq targets separately, consisting of all peaks detected at one or more time points. Reads overlapping these consensus peak sets were calculated and normalized using Trimmed Mean of M-values (TMM) method with control read counts subtracted and using full library sizes for normalization. The resulting ChIP-seq affinity data was used for principal component analysis using DiffBind functionality.

BEDTools [68] multiIntersectBed and merge tools were used for detecting peak overlaps between the three time point samples, retaining the information on the peak existence in a sample, and for merging the overlapping peak intervals into continuous peak regions. TMM normalized read counts for the merged peaks were obtained using DiffBind. Log2 fold change

of normalized read counts between day 7 and day 0 samples, day 14 and day 0 samples as well as day 14 and day 7 samples were calculated.

HOMER (Hypergeometric Optimization of Motif EnRichment) software [69] was used for analyzing enrichment of known transcription factor (TF) motifs in ChIP-seq peaks and annotating the peaks with the known motifs.

Gene set enrichment of Gene Ontology Biological Process (GOBP) terms and Kyoto Encyclopedia of Genes and Genomes (KEGG) pathway terms was performed using R package CHiP-Enrich [70] with the gene locus definition set to 1 kb.

Integration of differential expression (DE) analysis data and ChIP-seq data was done using custom Rscripts and package ggplot2 [71] for visualization.

## Western blotting and co-immunoprecipitation

MC3T3-E1 cells were cultured on 10 cm dishes in maintenance medium or osteogenic medium. Protein lysates were collected from preconfluent, confluent, 7, 10 or 14 days differentiated cells after 10 minute incubation on ice in mRIPA lysis buffer (50mM Tris-HCl, pH 7.4; 0.5% NP-40; 0.25% sodium deoxycholate; 150mM sodium chloride; 1 mM sodium orthovanadate; 0.5 mM phenylmethylsulfonyl fluoride) supplemented with Pierce Mini protease inhibitors (Thermo Scientific). Cell lysates were homogenized by purging through a 27G needle and cell debris discarded by centrifuging for 15 minutes at 16000G at 4°C. Protein concentration was measured using Bradford Protein Assay (Bio-Rad).

Protein samples were separated using 4–20% gradient SDS-PAGE gels (Bio-Rad Mini-PROTEAN TGX) and transferred to 0.22μm nitrocellulose membranes (Maine Manufacturing LLC). Membranes were blocked with either 5% non-fat dried milk in TBST (150mM NaCl, 0.05% Tween20, 10mM Tris pH 7.5) and incubated with primary antibodies against H3, H3K4me1, H3K4me2, H3K9me1, H3K9me2 or with 5% BSA in TBST and incubated with anti-Lsd1 primary antibody. Horseradish peroxidase conjugated anti–mouse (Lsd1) or anti-rabbit (other antibodies) IgG was used as a secondary antibody. Proteins were visualized using WesternBright Quantum detection kit (Advansta). Antibody information can be found in S1 Table.

## Lsd1 inhibition *in vivo*

Mouse studies were approved by the Finnish ethical committee for experimental animals (license numbers 5186/04.10.07/2017 and 14044/2020), complying with the international guidelines on the care and use of laboratory animals. Ten 6-week-old C57BL/6NHsd male mice were treated with either 0.9% NaCl (vehicle) or 1.5mg/kg GSK-LSD1 delivered by intraperitoneal injections in a 4-on-3-off schedule for four weeks. The weight of the animals was monitored biweekly and the overall wellbeing five times a week. The mice were injected intraperitoneally with calcein (20 mg/kg) and demeclocycline (40 mg/kg) (both from Sigma-Aldrich) 7 and 2 days prior to sacrifice, respectively. The mice were euthanized with $CO_2$-asphyxiation followed by blood sample collection via cardiac puncture. Bone samples for μCT and histomorphometry were fixed with 10% Neutral Formalin Buffer overnight and transferred to and stored in 70% ethanol.

## Lsd1 conditional knockout mice

Mice embryos carrying the floxed Kdm1a/Lsd1 alleles (Lsd1$^{fl/fl}$, stock #023969) were obtained from the Jackson laboratory. Lsd1$^{fl/fl}$ mice were crossed with Prrx1-Cre to create limb bud mesenchyme specific homozygous and heterozygous conditional Lsd1-knockout mice (Lsd1$_{Prrx1}^{-/-}$, Lsd1$_{Prrx1}^{+/-}$ respectively). The mice were injected intraperitoneally with calcein (20

mg/kg) and demeclocycline (40 mg/kg) (both from Sigma-Aldrich) 3 and 1 days prior to sacrifice, respectively. Histological samples were collected at 4 weeks of age as described previously. Marrow-evacuated femurs were snap-frozen in liquid nitrogen and stored in -80˚C for RNA extraction. Descriptive 2D X-ray images of the limbs were taken ex vivo using a MX-20 imaging device (Faxitron, USA).

## Histological analysis

EDTA-decalcified tibias were embedded in paraffin and 4-μm thick sections were collected. Safranin-O staining was done using conventional staining protocols and imaged using the Pannoramic 250 slide scanner (3d Histech). Collagen was visualized using the Picro-Sirius Red Stain Kit for Collagen (Biosite), imaged using 90˚ polarized light visualizing thin collagen fibers as green and thick collagen fibers as yellow-red, and analyzed using FIJI software.

For growth plate analysis, HE stained sagittal sections of proximal tibia epiphysis were used. Average thickness of one medial and two marginal lines of the proliferative and hypertrophic growth plate (GP) zones were measured by ImageJ. Cell density of the proliferative and hypertrophic zones of proximal tibia were analyzed by counting average cell number in 8 squares (each 10,000 $\mu m^2$ in size) covering the entire zone. Cell column heights were measured from 10 representative columns in proliferative and hypertrophic zones across the growth plate.

## X-ray microcomputed tomography analysis

Femurs of the mice treated with Lsd1 inhibitor GSK-LSD1 were cleaned of soft tissue and analyzed with X-ray microcomputed tomography (μCT). Cortical and trabecular bone structure of the distal femurs were analyzed using μCT (SkyScan 1070, Kontich Belgium) with 8,37μm resolution. After scanning, images were reconstructed (NRecon 1.4, SkyScan) and reoriented to ensure comparability (Dataviewer, SkyScan). The regions of interest (ROI) were drawn (CTan 1.4.4, SkyScan) on every tenth layer from 120 layers, with a total height of ~1000μm for trabecular bone, and from 100 layers with a total height of ~840μm for cortical bone and the results were then quantified and analyzed. The ROIs were drawn blinded for the treatment groups and quantified.

Tibias of the Lsd1cKO-mice were cleaned of soft tissue and their trabecular and cortical bone structures were analyzed using 5μm resolution (Skyscan 1074, Kontich Belgium). Upon scanning, the reconstruction (NRecon 1.7.5.6, SkyScan) and reorientation of images were done as described above. The regions of interest (ROI) were drawn on every tenth layer from 100 layers 210 layers (1050μm) down from the distal end of the proximal growth plate for the trabecular bone, and 1000 layers (5000μm) down for the cortical bone and subsequently analyzed (CTan 1.19.10.2, SkyScan). The ROIs were drawn blinded for all genotypes and the data was then quantified.

## Histomorphometric analysis

After fixation, tibias were embedded in methyl methacrylate (MMA) (Sigma-Aldrich, USA). 5μm sagittal sections were cut using a Leica RM2165 rotatory microtome and von Kossa, Toluidine blue and Tartrate-resistant acid phosphatase (TRACP)-stainings were performed on deplastified sections with standard protocols. In brief, TRACP-staining visualizes osteoclasts using osteoclast-specific acid phosphatase activity. The slides were analyzed using Osteomeasure-histomorphometry workstation (OsteomeasureXP 3.1.0.1, Osteometrics, USA). The analyzed area of the tibias was defined as 1,30 mm x 0,84 mm, starting 200μm distally from the growth plate, excluding the cortical border areas with a 100μm margin. Static parameters were

measured from toluidine blue and TRACP stained slides and dynamic parameters from unstained slides according to standardized protocols [72].

Subepiphyseal TRACP-staining and bone area were measured from an area defined as 1,00 mm x 0,20 mm starting immediately distally from the proximal growth plate of the tibia using FIJI.

### Bone turnover markers

Blood samples were collected at sacrifice via cardiac puncture. Blood samples were allowed to clot and serum was collected after centrifugation and stored at -80˚C. Bone formation and resorption were assessed using serum levels of N-terminal propeptides of type I collagen (P1NP) and C-terminal telopeptides of type I collagen (CTX-1) using Rat/Mouse PINP EIA and RatLaps CTX-I EIA kits (IDS, UK), respectively.

### Statistical analysis

Results are presented as mean with standard deviation (SD). The results were tested for normal distribution and outliers using suitable tests and changes between groups were analyzed using Student's two-tailed T-test, one-way ANOVA or Kruskall-Wallis test using Bonferroni's multiple comparison correction where applicable with the GraphPad Prism 8.3.0. Statistical significance was set to $p < 0,05$.

## Results

### Lsd1 is abundantly expressed in mouse osteoblastic cells

We performed whole transcriptome sequencing of MC3T3-E1 cells at different stages of differentiation, ranging from proliferative cells to mature osteoblasts. Progression of differentiation was verified by alkaline phosphatase (*ALP*) activity and formation of mineralized nodules at the end of the 21-day culture (Fig 1A). Successful osteoblast differentiation was also confirmed by the expression of osteoblast-related genes in the RNA-seq data set (Fig 1B). In the expression analyses of different epigenetic regulators, we found relatively abundant expression of several *Kdm*s (Fig 1C). Interestingly, there were significant changes in *Kdm* expression levels in mature osteoblasts at 21 days compared to proliferating cells (Table 1). Of the MAO family of KDMs, which is much less studied in osteoblasts than the larger JmjC-domain containing KDM family, *Kdm1a/Lsd1* expression was much more abundant than *Kdm1b/Lsd2* mRNA, suggesting a more prominent role for Lsd1 in these cells. Indeed, *Lsd1* was one of the most abundantly expressed Kdms in MC3T3-E1 cells throughout the 3-week differentiation time course (Fig 1C). We confirmed the robust expression of *Lsd1* mRNA in both MC3T3-E1 and mouse primary calvarial cells by quantitative RT-PCR but did not did not observe significant changes in the mRNA expression levels during osteoblast differentiation. Lsd1 protein level on the other hand showed an increase in the differentiated osteoblasts compared to the proliferating cells, suggesting stabilization of Lsd1 protein in mature osteoblasts (S1 Fig). The relatively stable expression of *Lsd1/Kdm1a* in mouse osteoblasts is consistent with previous observations in human MSCs [15].

### Lsd1 downregulation inhibits osteogenic differentiation

The prominent expression of Lsd1 in osteoblasts and its previously reported role in regulating adipocyte differentiation prompted us to test whether Lsd1 controls osteoblast differentiation and function. We first silenced *Lsd1* expression in MC3T3-E1 cells by stable expression of Lsd1 shRNAs. Of the five shRNAs we tested, two shRNA constructs (shLSD1-73 and shLSD1-76) showed approximately 50% downregulation of *Lsd1* mRNA levels (Fig 1D). Lsd1

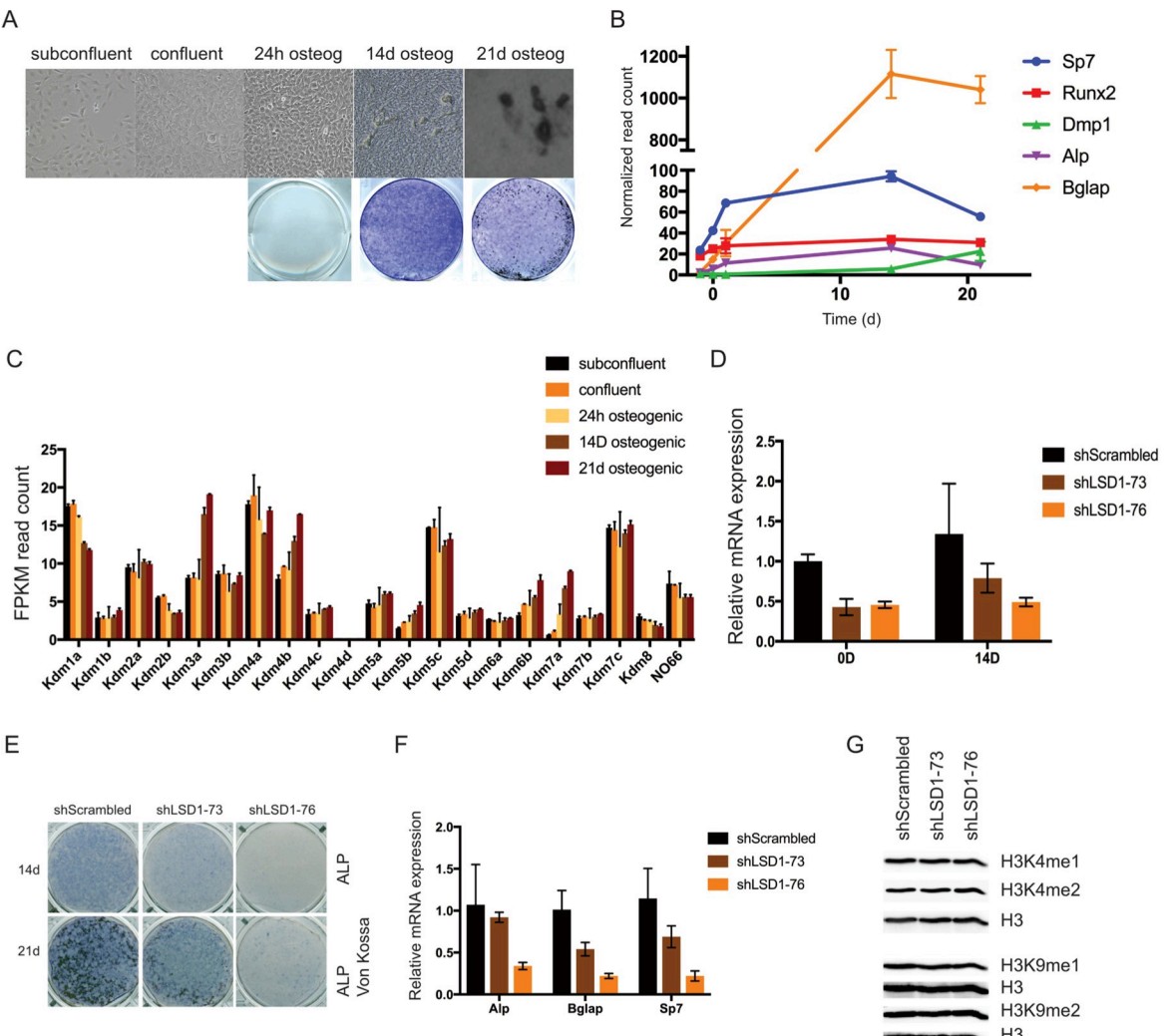

**Fig 1. Lsd1 is abundantly expressed in osteoblasts and important for osteoblast differentiation.** (A, B) RNA-seq analysis was performed from MC3T3-E1 at different stages of differentiation. The differentiation was verified by Von Kossa (top) and ALP staining (bottom) (A) and by expression profiles of osteoblast biomarkers in RNA-seq data (n = 2)(B). (C) mRNA expression of *KDMs* in MC3T3-E1 cells at different stages of differentiation measured by RNA-seq. Average of FPKM normalized read counts from two replicate samples for each time point are shown. (D) Knockdown of *Lsd1* mRNA in MC3T3-E1 cells by stable shRNA expression shown by quantitative qPCR. B-actin normalized *Lsd1* mRNA expression relative to shScrambled cells from three replicate samples is shown. (E) shLSD1 knockdown cell and control shScrambled were cultured in standard osteogenic conditions for 14 and 21d and fixed in 10% formalin. Differentiated phenotype was demonstrated by ALP and Von Kossa staining and by (F) quantitation of osteogenic genes mRNA expression at 14D differentiated cells by qPCR (n = 3). (G) Western blot showing Lsd1 protein and methylation state of histone H3 in shLSD1 knockdown cells. (E, F) Results from a representative cell culture experiment from three independent experiments is shown. P-values for statistically significant differences are marked * P<0.05, ** P<0.01, *** P<0.001.

knockdown impaired osteoblast differentiation as demonstrated by reduced ALP activity and formation of mineralized nodules (Fig 1E), as well as decreased expression of osteoblast related genes (Fig 1F), suggesting an important role for Lsd1 in the regulation of osteogenic differentiation and function. Because Lsd1 is a histone demethylase, we furthermore evaluated global methylation levels of H3K4 or H3K9 in these cells. Lsd1 downregulation did not affect overall H3K4/H3K9 methylation status (Fig 1G). This important observation indicates that reduced levels of Lsd1 may interfere with osteoblastogenesis primarily through locus-specific chromatin mechanisms rather than effects on bulk chromatin.

**Table 1. More than two-fold differentially expressed histone demethylases in MC3T3-E1 cells differentiated for 21 days compared to undifferentiated subconfluent cells in RNA-seq analysis.**

| Histone demethylase | Synonyms | Fold change | Adjusted p-value |
|---|---|---|---|
| **Kdm3a** | Jhdm2a, Jmjd1a, Kiaa0742 | 2,4 | 7.36e-12 |
| **Kdm4b** | Jhdm3b, Jmjd2b | 2,2 | 5.87e-15 |
| **Kdm5b** | Jarid1b, Kiaa4034, Plu1 | 3,1 | 2.11e-20 |
| **Kdm6b** | Jmjd3, Kiaa0346 | 2,5 | 1.39e-07 |
| **Kdm7a** | Jhdm1d, Kdm7, Kiaa1718 | 13,4 | 3.05e-64 |

## Pharmacological inhibition of Lsd1 suppresses osteogenic differentiation in vitro

To test whether the effect of Lsd1 on osteoblast differentiation depends on its enzymatic activity, we treated MC3T3-E1 cells with selective Lsd1 inhibitors RN-1 and GSK2780854A (GSK-LSD1). RN-1 is a tranylcypromine (TCP) derivative with reported IC50 value of 70 nM for Lsd1 inhibition [73], while GSK-LSD1 has an IC50 of 16 nM and high specificity for Lsd1 over MAOs [74]. RN-1 suppressed osteoblast differentiation as shown by reduced ALP staining and expression of osteoblast-related genes at nanomolar concentrations (Fig 2A). Similarly, GSK-LSD1 impaired osteoblast differentiation at all concentrations tested (Fig 2B), with no observed effects on cell viability. Importantly, both RN-1 and GSK-LSD1 completely inhibited the formation of mineralized nodules also in mouse primary calvarial cell cultures, although their effects on ALP activity were less pronounced than in MC3T3-E1 cells (Fig 2C). Because no inhibition of other enzymes has been reported at these low concentrations [73, 74], our results suggest that RN-1 and GSK-LSD1 suppress osteoblast differentiation due to the specific inhibition of Lsd1 activity.

## Lsd1 inhibition suppresses osteoblast function *in vivo*

To study the effect of selective Lsd1 inhibition on bone metabolism *in vivo*, we treated 6-week-old wild-type C57BL/6NHsd male mice with either 1,5 mg/kg GSK-LSD1 or vehicle for 4 weeks intraperitoneally (n = 10/group). The mice tolerated the treatment well with no acute side effects. Two mice treated with GSK-LSD1 developed an eye infection, but this was most likely not related to the treatment and the mice healed normally with sterile saline rinsing.

Assessment of the distal femora by μCT revealed that trabecular bone was significantly decreased in GSK-LSD1 treated mice compared to vehicle treated mice, as shown by reduced bone volume (BV/TV), trabecular thickness and trabecular number (Fig 3A and 3B). Cortical thickness and cortical bone volume were also decreased in GSK-LSD1 treated mice (Fig 3C).

To investigate the underlying mechanism for the decreased trabecular bone parameters, we performed histomorphometric analysis on the methyl methacrylate-embedded tibias. Histomorphometry confirmed the trabecular osteopenia observed by μCT (Fig 3D). This was due to decreased number and activity of the osteoblasts as the number of osteoblasts, osteoid surface and thickness, as well as bone formation rate were all decreased in GSK-LSD1 treated mice compared to vehicle treated controls. There were no significant differences in any of the osteoclast parameters (Fig 3D–3G).

To further elucidate the effects of Lsd1 inhibition on bone turnover rate, we measured the levels of bone resorption marker CTX-I and bone formation marker P1NP in serum samples. There was no significant difference in serum CTX-I (p = 0.67), while serum P1NP was significantly decreased (p = 0.012), supporting the histomorphometric finding of decreased bone formation (Fig 3H).

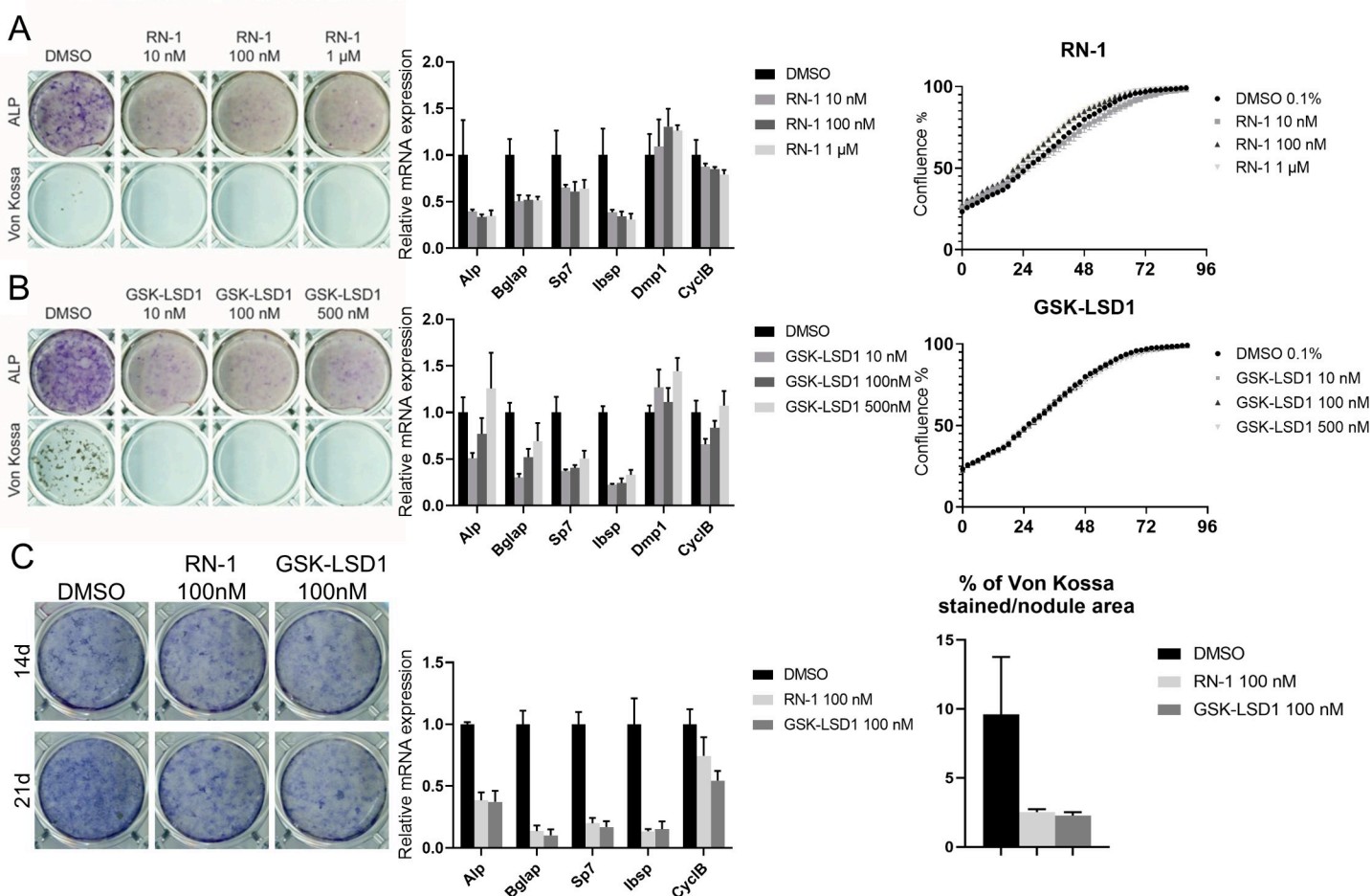

**Fig 2. Lsd1 inhibition in osteoblasts.** Effect of Lsd1 inhibition in MC3T3-E1 cells by RN-1 (A) and GSK-LSD1 (B) on differentiation demonstrated by ALP and Von Kossa staining of 21d differentiated cells (left), expression of osteoblast related genes in 14d differentiated cells quantitated by qPCR (middle, n = 3) and proliferation rate (right, n = 3). Effect of Lsd1 inhibition in calvarial osteoblasts by RN-1 and GSK-LSD1 (C) demonstrated by ALP staining of 21d differentiated cells (left), expression of osteoblast related genes in 14d differentiated cells quantitated by qPCR (middle, n = 3) and von Kossa stained area (right, n = 3). P-values for statistically significant differences are marked * P<0.05, ** P<0.01, *** P<0.001.

## Loss of Lsd1 expression disrupts primary spongiosa ossification and reorganization

To study the loss of Lsd1 expression in long bone osteoblasts we crossed the Lsd1$^{fl/fl}$ mice with Prrx1-Cre mice, leading to deletion of *Kdm1a* exons 5 and 6 in the limb bud mesenchymal cells [51]. The male Lsd1$_{Prrx1}^{-/-}$ mice showed a clear bone phenotype already at 4 weeks of age with 42% lower body weight (p<0.0001) and 35% shorter tibias (p<0.0001) compared to littermate Lsd1$^{fl/fl}$ controls. The heterozygous Lsd1$_{Prrx1}^{+/-}$ showed a less pronounced phenotype with 10% lower body weight and 5% shorter tibias compared to littermate Lsd1$^{fl/fl}$ controls (Fig 4A). Due to the drastic phenotype the bone samples were collected at 4 weeks of age to avoid potential harm for animal wellbeing.

In visual analysis of the 2D X-ray data the long bone phenotype of Lsd1$_{Prrx1}^{-/-}$ mice was apparent, with osteogenesis imperfecta-like curvature of the bones, loss of secondary ossification centres and impaired union of the fibula to the tibia (Fig 4B). The μCT 3D reconstructions showed unclosed calvarial sutures of the Lsd1$_{Prrx1}^{-/-}$ mice and similar phenotype of tibias as the

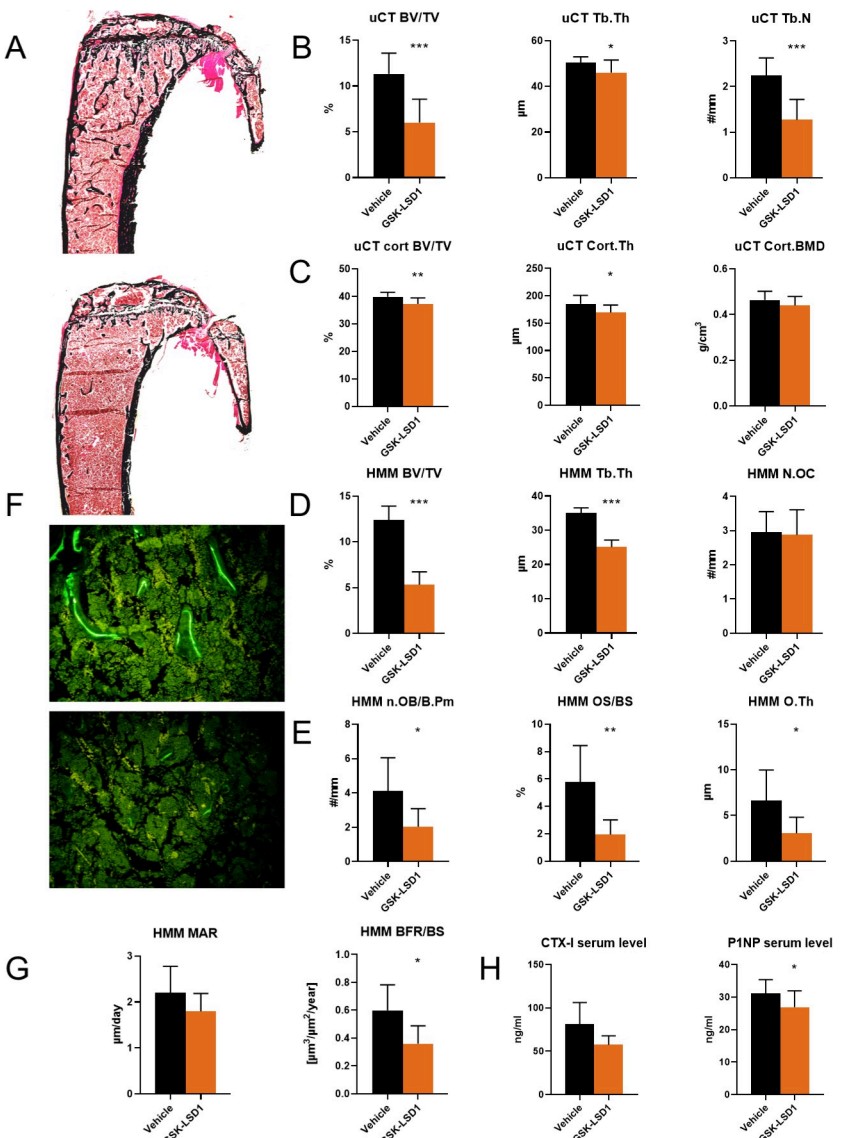

**Fig 3. Lsd1 inhibitor treatment in vivo.** (A) Representative images of Von Kossa stained control (top) and GSK-LSD1 treated (bottom) MMA embedded sections. (B) µCT analysis of trabecular bone. (C) µCT analysis of cortical bone. (D) Histomorphometric analysis of trabecular bone structural parameters. (E) Histomorphometric analysis of osteoblast parameters. (F) Representative images of fluorescent labels in control (top) and GSK-LSD1 treated (bottom) samples. (G) Histomorphometric analysis of dynamic parameters, (H) Bone resorption marker CTX-I and bone formation marker P1NP serum levels. Mice were treated with either 1,5mg/kg GSK-LSD1 or saline on a 4-on-3-off schedule for 4 weeks, n = 10/group). P-values for statistically significant differences are marked * P<0.05, ** P<0.01, *** P<0.001.

initial 2D images (Fig 4C). Further 3D analysis showed significantly decreased trabecular BV/TV and number as well as increased structural model index (SMI) in the $Lsd1^{-/-}_{Prrx1}$ mice (Fig 4D), indicating loss of structural complexity and remodeling of the trabecular bone. In the analysis of cortical bone at tibial midshaft of the $Lsd1^{-/-}_{Prrx1}$ mice, the cortical volume and cortical BMD were both decreased, while cortical porosity was increased when compared to controls (Fig 4D). Histomorphometric analysis of the tibias from $Lsd1^{-/-}_{Prrx1}$ mice showed similarly lower overall tissue volume, consistent with genetic effects on limb tissue patterning, but increased BV/TV and trabecular thickness indicating enhanced bone formation. Osteoblast parameters

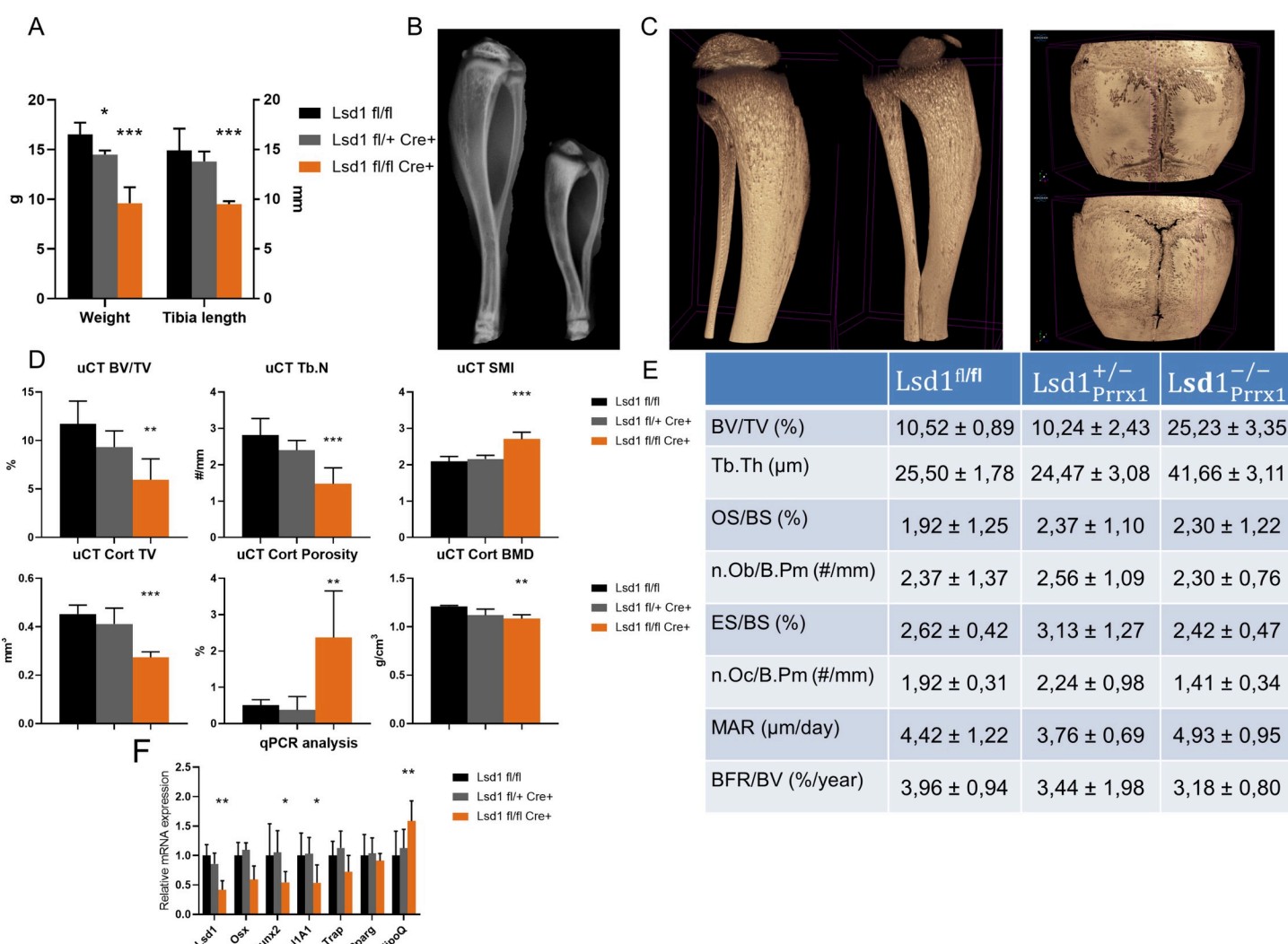

**Fig 4. Loss of Lsd1 in long bone mesenchymal cells leads to impaired ossification and growth.** (A) Lsd1 knockout mice have decreased body weight and length of tibia. (B) 2D X-ray data shows osteogenesis imperfecta-like phenotype with increased curvature, impaired secondary ossification and non-union of the fibula to the tibia in the Lsd1-cKO mice (right) compared to control (left). (C) Representative 3D models of μCT data shows similar loss of secondary ossification site and unclosed calvarial sutures in the Lsd1-cKO mice (right, bottom) compared to the control mice (left, top). (D) 3D analysis of trabecular and cortical bone. (E) Histomorphometric analysis of trabecular bone. (F) RT-PCR analysis of RNA extracted from marrow-evacuated bone shows efficiency of *Lsd1* knockout (~60%), decrease in osteoblast markers *Osx*, *Runx2* and *Col1A1*, no significant change of osteoblast marker *Trap* and adipocyte marker *PPARgamma*, and a significant increase of *AdipoQ* expression. A-E: n = 5-7/group, F: n = 5/group. P-values for statistically significant differences are marked * P<0.05, ** P<0.01, *** P<0.001.

were similar between groups, but the number of osteoclasts decreased in $Lsd1_{Prrx1}^{-/-}$ mice, although it did not reach statistical significance (p = 0.36). There were no differences in the dynamic markers of bone formation between $Lsd1_{Prrx1}^{-/-}$ mice and controls (Fig 4E). These changes in imaging parameters indicate that genetic loss of Lsd1 function in the mesenchyme has complex effects on bone formation.

Gene expression analysis on RNA extracted from marrow-evacuated femurs showed a 60% decrease of *Lsd1* expression in the knockout mice compared to controls, confirming the effect of the conditional knockout model. Decreased expression was also found on osteoblast markers Osterix/Sp7 (*Osx*, 41%, p = 0.060), Osteoprotegerin (*Opg/Tnfrs11b*, 52%, p = 0.051), osteoblast master regulator Runx2 (45%, p = 0.032) and Collagen type 1 alpha 1 (*Col1A1*, 46%,

p = 0.029). Osteoclast marker *Trap/Acp5* expression was decreased by 27% but the change was not statistically significant. Adipocyte master regulator Ppargamma/*Pparg* expression was unchanged but adiponectin (AdipoQ/*Adipoq*) expression was increased by 59% (p = 0.0043) (Fig 4F). Expression values were normalized to beta-actin/*Actb* and compared between $\text{Lsd1}^{-/-}_{\text{Prrx1}}$ and $\text{Lsd1}^{\text{fl/fl}}$ mice.

Histological analysis of decalcified tibias verified the aberrations in the growth plate first observed in the radiological analysis of X-ray data (Fig 4B). Visual analysis of Safranin-O staining showed impaired organization and decreased cellularity of the growth plate. The proliferative zone of the growth plate was also significantly thinner in $\text{Lsd1}^{-/-}_{\text{Prrx1}}$ mice, unlike the hypertrophic zone (Fig 5A). In the growth plate analysis, the chondrocyte density of both the proliferative and hypertrophic zones were decreased (Fig 5B). The mean chondrocyte column height in both the proliferative zone and hypertrophic zone were decreased in $\text{Lsd1}^{-/-}_{\text{Prrx1}}$ mice (Fig 5C). The Picrosirius Red staining of trabecular bone combined with polarized light imaging (which discriminates between collagen isoforms) revealed increased Collagen I (Col1a1) staining and decreased Collagen III (Col3a1) staining when normalized to bone area. There were no changes in the collagen composition of cortical bone of $\text{Lsd1}^{-/-}_{\text{Prrx1}}$ mice (S2 Fig). The subepiphyseal osteoclast count showed a decreased number of osteoclasts normalized to bone area (Fig 5D). These results indicate that the growth plates of $\text{Lsd1}^{-/-}_{\text{Prrx1}}$ mice exhibit loss of organization and cellular richness, as well as loss of remodeling of subepiphyseal bone.

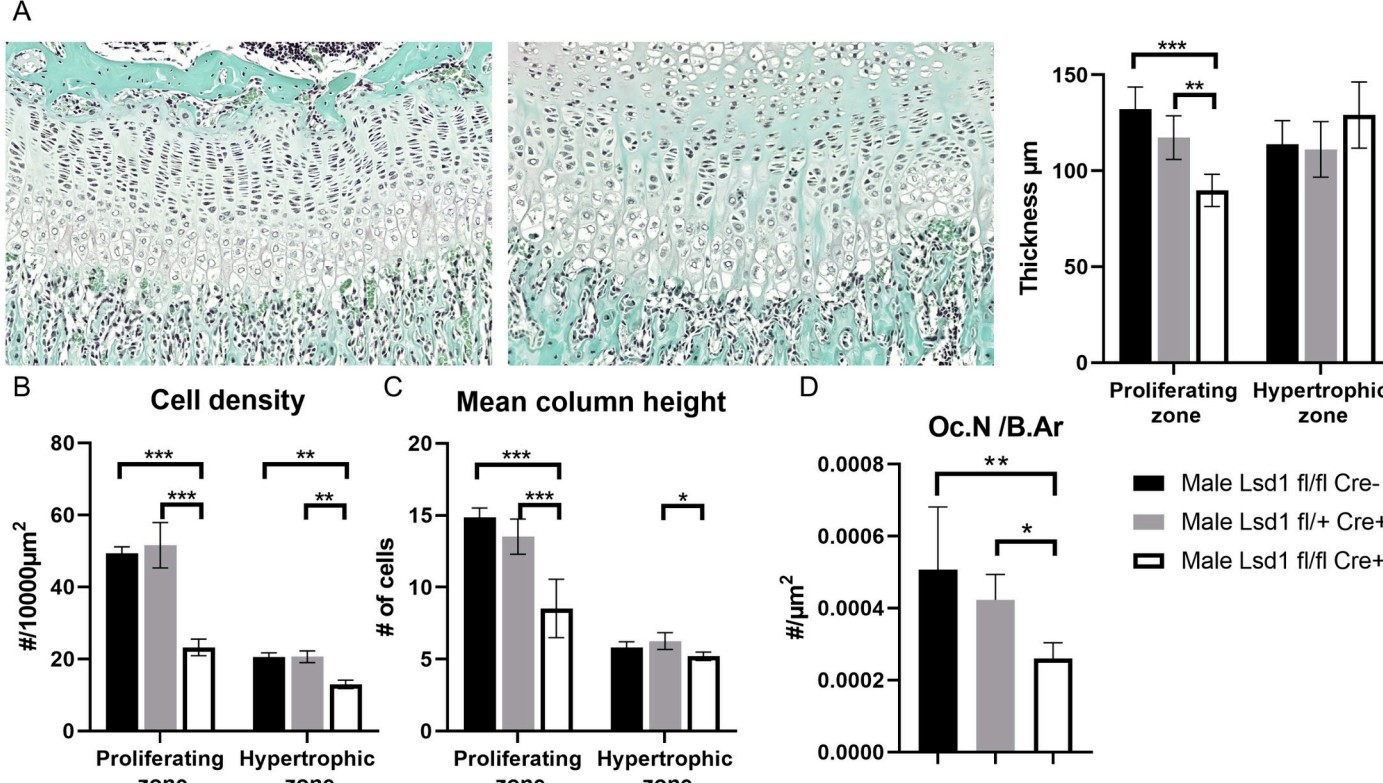

**Fig 5. Loss of Lsd1 in long bone mesenchymal cells disrupts growth plate organization but not collagen maturation.** (A) Growth plates of Lsd1 knockout mice have decreased organization and thicker hypertrophic zone (right) compared to control (left) (n = 4–5 per group). (B) Mean cell density of Lsd1 knockout mice was lower in both proliferating zone and hypertrophic zone of the growth plate. (n = 4–5 per group). (C) Chondrocyte column height counting showed impaired cellular organization in the proliferating zone of growth plate. (n = 4–5 per group). (D) Osteoclast counting of subepiphyseal bone showed impaired regeneration seen as decreased relative number of osteoclasts (n = 5 per group).

## Lsd1 binding is enriched on proximal promoter areas during osteoblastogenesis

To study the genomic occupancy of Lsd1 and the dynamics of histone methylation during osteoblast differentiation, we performed ChIP-seq analysis of Lsd1, H3K4me1, H3K4me2, H3K4me3 and H4K27me3 on differentiated MC3T3-E1 cells at 0,7 and 14 days. In the Lsd1 ChIP-seq data, we found 9207, 10879 and 16978 significant Lsd1 peaks compared to input control in cells differentiated for 0, 7, and 14 days, respectively, suggesting that Lsd1 occupancy increases upon osteoblastic differentiation (Fig 6A). There were 3755 common peaks present in all time points, representing constitutive Lsd1 binding sites, but also a large number of unique Lsd1-bound sites at various time points (e.g., more than 9,000 unique binding sites in 14D differentiated cells), demonstrating dynamic binding of Lsd1 in osteoblast genome during differentiation (Fig 6B). Interestingly, Lsd1 binding showed strong enrichment at transcriptional start sites (TSS) and the proximal promoters as the majority of the Lsd1 peaks were localized within 1 kb of TSS, and this peak region narrowed further upon osteoblast differentiation (Fig 6C and 6D). This finding suggests that Lsd1 is modulating gene expression by acting directly on TSS and on proximal promoters.

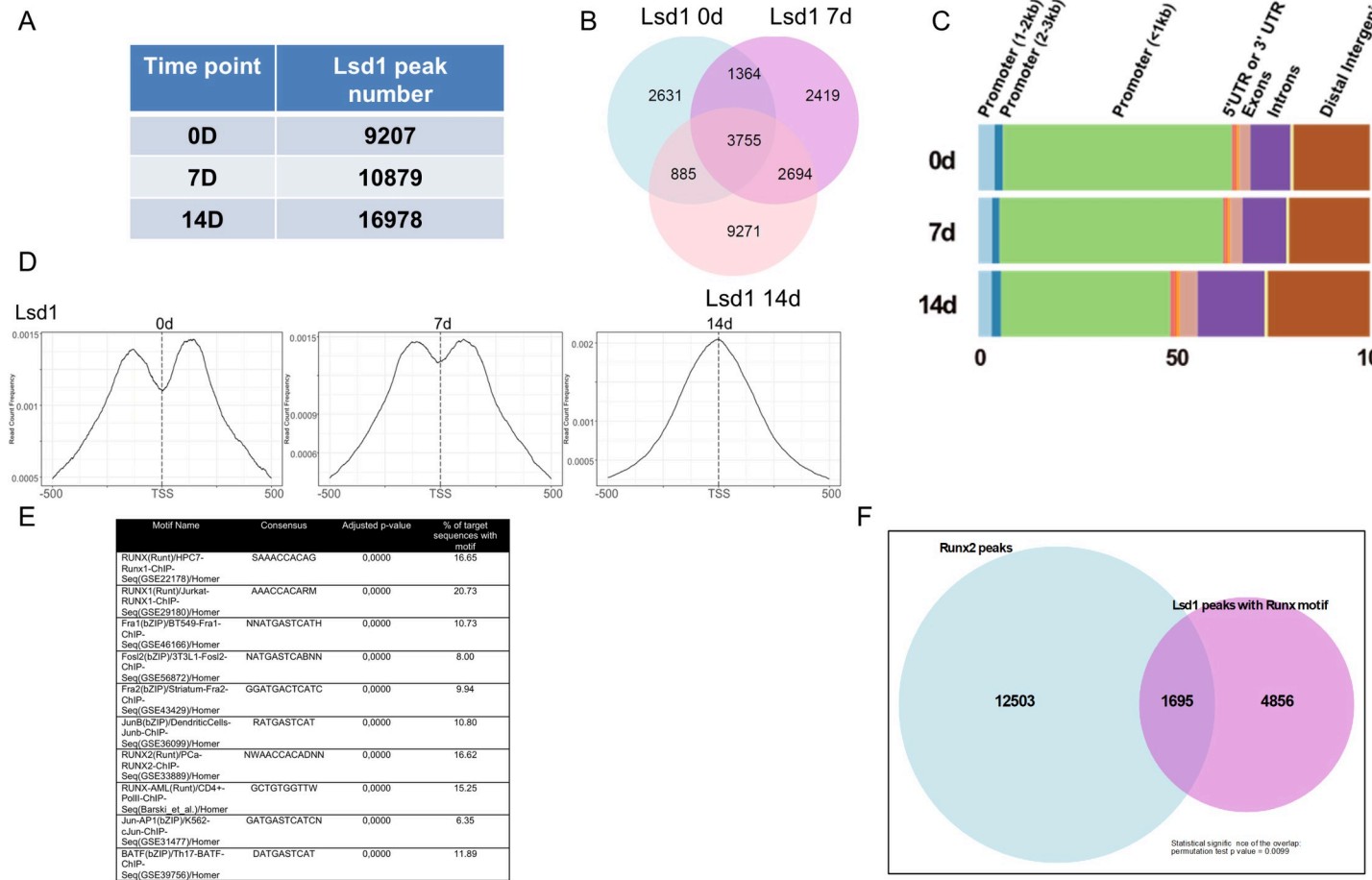

**Fig 6. Lsd1 ChIP-seq analysis.** (A) Number of Lsd1 peaks detected at different time points. (B) Overlap of Lsd1 peaks between different time points represened by Venn diagram. (C) Distrubution of Lsd1 peaks. (D) Distance of Lsd1 peaks from the TSS at different time points. (E) Top enriched TF consensus motifs in Lsd1 ChIP-seq peak data. (F) Overlap of Lsd1 and published Runx2 ChIP-seq peaks.

Altogether, 5522, 6557 and 9319 genes were bound by Lsd1 on days 0, 7 and 14 of osteogenic differentiation, respectively. Between 1 and 33 Lsd1 peaks were identified on these genes, while the average gene was occupied by 1.7 to 1.8 Lsd1 peaks at each time-point. Based on these data, Lsd1 likely plays a prominent role in the regulation of osteogenesis by acting at gene promoters. The full list of Lsd1 ChIP-seq peak locations and genes are presented in S2 Table.

## Lsd1 interacts with loci enriched for Runx2 binding motifs

Previous reports have demonstrated that Lsd1 binds to DNA, but in a non-specific manner [75–77]. However, we hypothesized that Lsd1 could interact with other nuclear proteins that would target Lsd1 to specific gene promoters to regulate gene expression in osteoblasts. Therefore, we searched for conserved transcription factor consensus binding motifs in Lsd1 bound sites using the HOMER tool. Intriguingly, in the top ten most enriched motifs at Lsd1 bound sites, we found consensus motifs for Runx1 and Runx2, and for different subunits of the AP-1 transcription factor complex, including Fra1, Fra2 and Fosl1 (Fig 6E). Consensus Runx1/2 motifs ( (A/T/C)TGTGGTT(A/T); (G/T)(T/C)TGTGGTTT; CTGTGGTTT(G/C) ) were found in 16–20% of Lsd1 ChIP-peaks. For comparison, Runx2 motifs are found in approximately 40% of Runx2 bound sites in MC3T3-E1 cells [78–80]. Strong enrichment of RUNX motifs in Lsd1 ChIP-seq peak data suggest that Lsd1 is important in modulating the promoter areas of osteoblast related genes allowing subsequent Runx2 binding or that Lsd1 interacts directly or indirectly with Runx2 in the same protein complex.

To further explore the putative functional interaction of Lsd1 with Runx2, we compared our Lsd1 ChIP-seq peak map with previously published Runx2 ChIP-seq data sets from MC3T3-E1 cells [78, 79]. Interestingly, of the Lsd1 bound sites that contained Runx2 binding motifs, 35% had been previously shown to bind Runx2, further supporting our interpretation of functional interplay between Lsd1 and Runx2 on the same sites on the chromatin in MC3T3-E1 osteoblasts (Fig 6F).

## Genomic distribution of Lsd1 dependent H3K4me marks with H3K27 methylation marks

In general, H3K4me1 is a marker for active enhancers, whereas H3K4me2 and H3K4me3 are marks for active promoters. H3K27me3 in turn, is present in silenced or poised/bivalent (marked by H3K4me3 and H3K27me3) chromatin [3]. Of these histone marks, H3K4me1 and H3K4me2 are substrates for Lsd1 demethylase activity. When evaluating peak numbers for these histone marks, the total peak numbers remained relatively unchanged between the differentiation time-points (Fig 7A). Consistent with the literature, we found H3K4me1 peaks more distant from the TSSs than H3K4me2/3, while H3K27me3 was nearly absent at TSSs but showed growing chromatin binding with increasing distance to the TSSs (Fig 7C). The distance of H3K4me2/3 from TSSs decreased upon differentiation, suggesting more condensed methylation of H3K4 at the core promoters in the mature osteoblasts. Binding of H3K27me3 in turn showed no marked dynamics between the time points. Accordingly, when the peak locations were correlated to annotated genomic features, most of H3K4me2/me3 were found at promoters, H3K4me1 at promoters and introns, and H3K27me3 both at promoters and distally in the intergenic regions (Fig 7B). In summary, the histone ChIP-seq peak data were consistent with the general characteristics described for the specific histone modifications we analyzed.

## Epigenomic correlation between Lsd1 and histone methylation status

To identify which histone marks are associated with Lsd1, we next studied the correlation between Lsd1 and histone mark peaks. In general, Lsd1 peak location at the TSSs was inversely

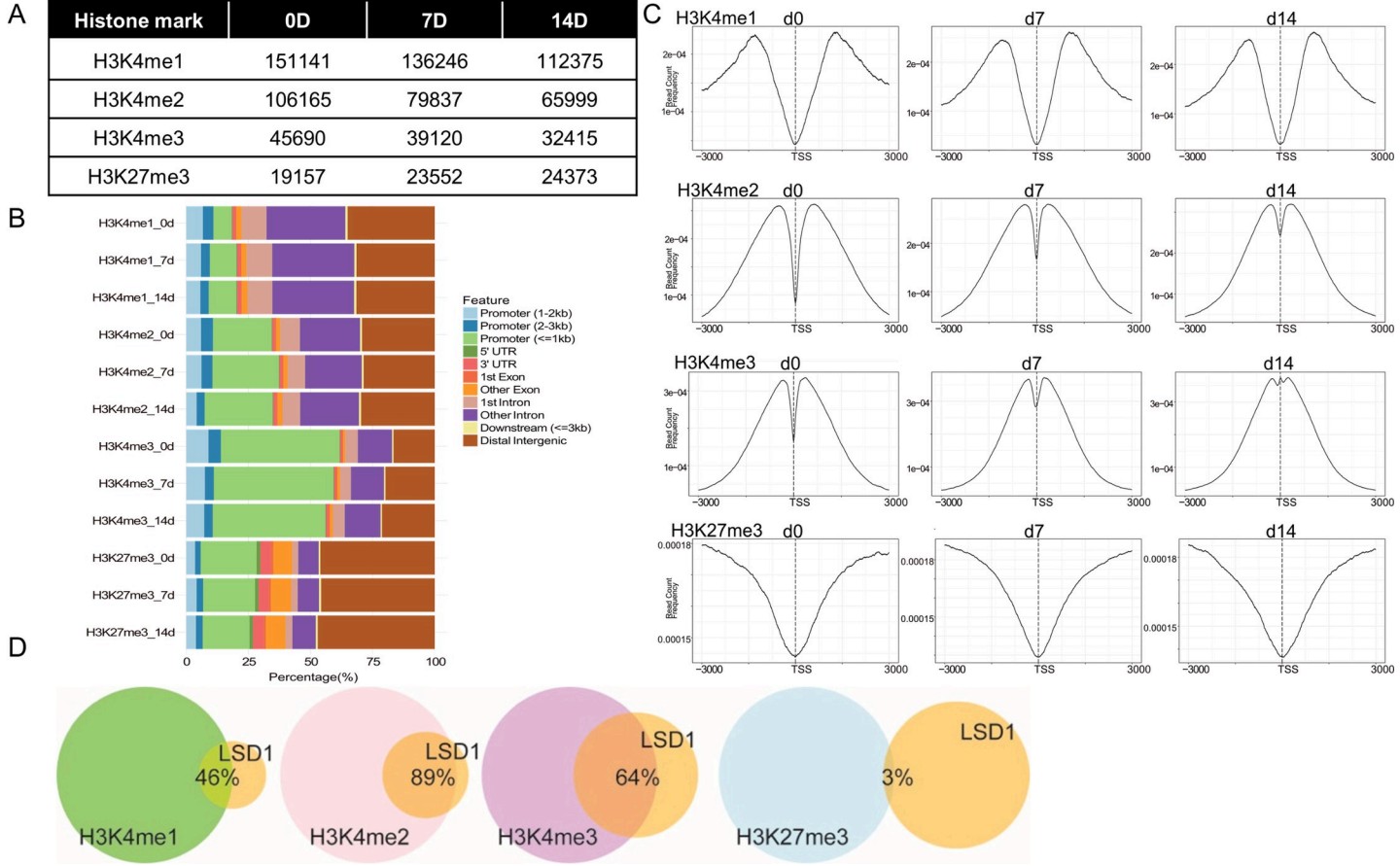

**Fig 7. Histone ChIP-seq analysis.** (A) Total number of ChIP-seq peaks. (B) Distribution of ChIP-seq peals on annotated genomic locations. (C) Average distance of H3K4me1/me2/m3 and H3K27me3 peaks from the TSS's. (D) Overlapping peaks of Lsd1 and histone mark binding sites at 14D differentiated cells. Percentage of Lsd1 peaks overlapping with histone modification peaks are indicated.

correlated with H3K4me1 occupancy, suggesting Lsd1 plays a role in demethylating H3K4me1 at the proximal promoters (Figs 6D and 7C). Lsd1, H3K4me2 and H3K4me3 in turn were found at or very close to the TSS. At many promoters, the peak profiles were almost identical between Lsd1 and H3K4me2 and H3K4me3, suggesting that Lsd1 associates with these two histone marks. In another approach, we evaluated the overlap of ChIP-seq peaks (Fig 7D and Table 2). On day 14 of osteogenic differentiation (day with highest number of Lsd1 occupied sites), 89% of the Lsd1 peaks overlapped with H3K4me2, whereas only 46% of the peaks overlapped with H3K4me1 peaks, both being known substrates for Lsd1. The difference in the peak overlap could suggest that in osteoblasts Lsd1 actively demethylates H3K4me1 leading to decreased H3K4me1 on the Lsd1 bound sites, whereas Lsd1 might not actively demethylate H3K4me2, although located at the same site. Alternatively, Lsd1 may bind H3K4me2 and upon de-methylation the

**Table 2. Overlapping LSD1 and histone modification peaks.**

| LSD1 peak number | Overlap with H3K4me1 | Overlap with H3K4me2 | Overlap with H3K4me3 | Overlap with H3K27me3 |
|---|---|---|---|---|
| **0D (8959)** | 30% | 89% | 76% | 2% |
| **7D (10574)** | 34% | 90% | 74% | 3% |
| **14D (16425)** | 46% | 89% | 64% | 3% |

enzyme is released from H3K4me1 marks, or that Lsd1 resides on active promoters near TSSs but participates in protein complexes not including H3K4me1 or me2. For comparison, only 2.5% of the Lsd1 peaks overlapped with H3K27me3 peaks, suggesting that Lsd1 is not present in the same protein complex that regulates H3K27me3 methylation state.

To study this in greater detail, Lsd1 peak locations at the proximal promoters of individual genomic loci were examined. In a view of chromosome 19 on our RNA-seq data (Fig 8), there are several constitutively expressed genes (except for two non-expressed genes marked by high H3K27me3). Interestingly, H3K4me1 was strongly decreased locally at Lsd1 peak regions at TSSs of actively transcribed genes, whereas there was abundant H3K4me2 binding at the same sites (Fig 8).

When Lsd1 and H3K4me1 were found at the same location apart from TSS, these sites were interpreted as enhancers. For example, for the Kdm7a locus, which is upregulated during osteoblast differentiation based on our RNA-seq data, we found that the Lsd1 peak correlated with low H3K4me1 at the promoter, but on an upstream enhancer, the Lsd1 peak site closely matched the local levels of H3K4me1 (S3 Fig). These examples demonstrate that the association of Lsd1 with H3K4me1 containing nucleosomes may be dependent on the genomic site. In other words, Lsd1 may act differentially on TSSs and enhancer areas. When we inspected promoters and enhancers of the individual osteogenic genes such as *Bglap*, *Sp7* and the locus containing both *Dmp1* and *Ibsp* (S4 Fig), we found that increased Lsd1 binding at 14d differentiated cells correlated with low H3K4me1 and increased H3K4me2 at these sites. These data

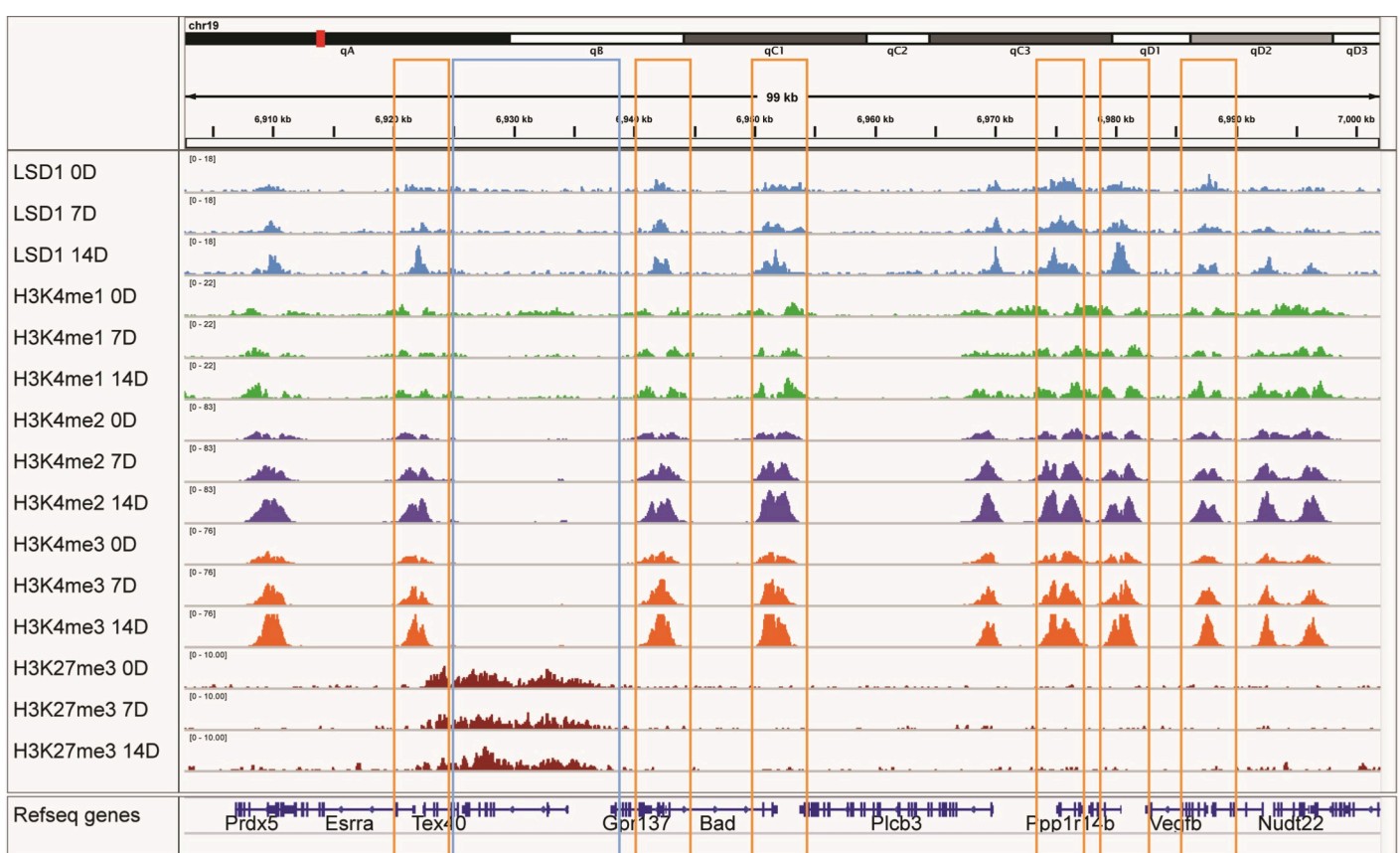

**Fig 8. Examples of H3K4 methylation patterns at Lsd1 bound promoters and enhancers visualized by integrative genomic viewer (IGV).** View of chromosome 19 with several expressed genes.

collectively suggest that Lsd1 may be involved in reorganizing chromatin of known osteogenic genes, and that Lsd1 may have substrate preference and perhaps preferentially demethylate H3K4me1 in this context.

## Lsd1 epigenomic events correlate with modulations in gene expression during MC3T3-E1 osteoblast differentiation

To understand the correlation between Lsd1 binding and transcription dynamics during osteoblast differentiation, we integrated our Lsd1 ChIP-seq data with our RNA-seq data of the parental MC3T3-E1 cells at 0 and 14 days of differentiation. First, we defined the gene as expressed if the log2 expression level was greater than 1 (Fig 9A). Intriguingly, when we studied the expression status of those genes occupied by Lsd1 within 1 kb from the TSS, we found that more than 90% of the genes were expressed (Fig 9B). Moreover, when including expression data for all 23,398 genes from the RNA-seq analysis, the average expression level of the genes occupied by Lsd1 was significantly higher than the expression level of non-Lsd1 occupied genes (Fig 9C). Thus, Lsd1 occupancy at the promoter was strongly associated with active

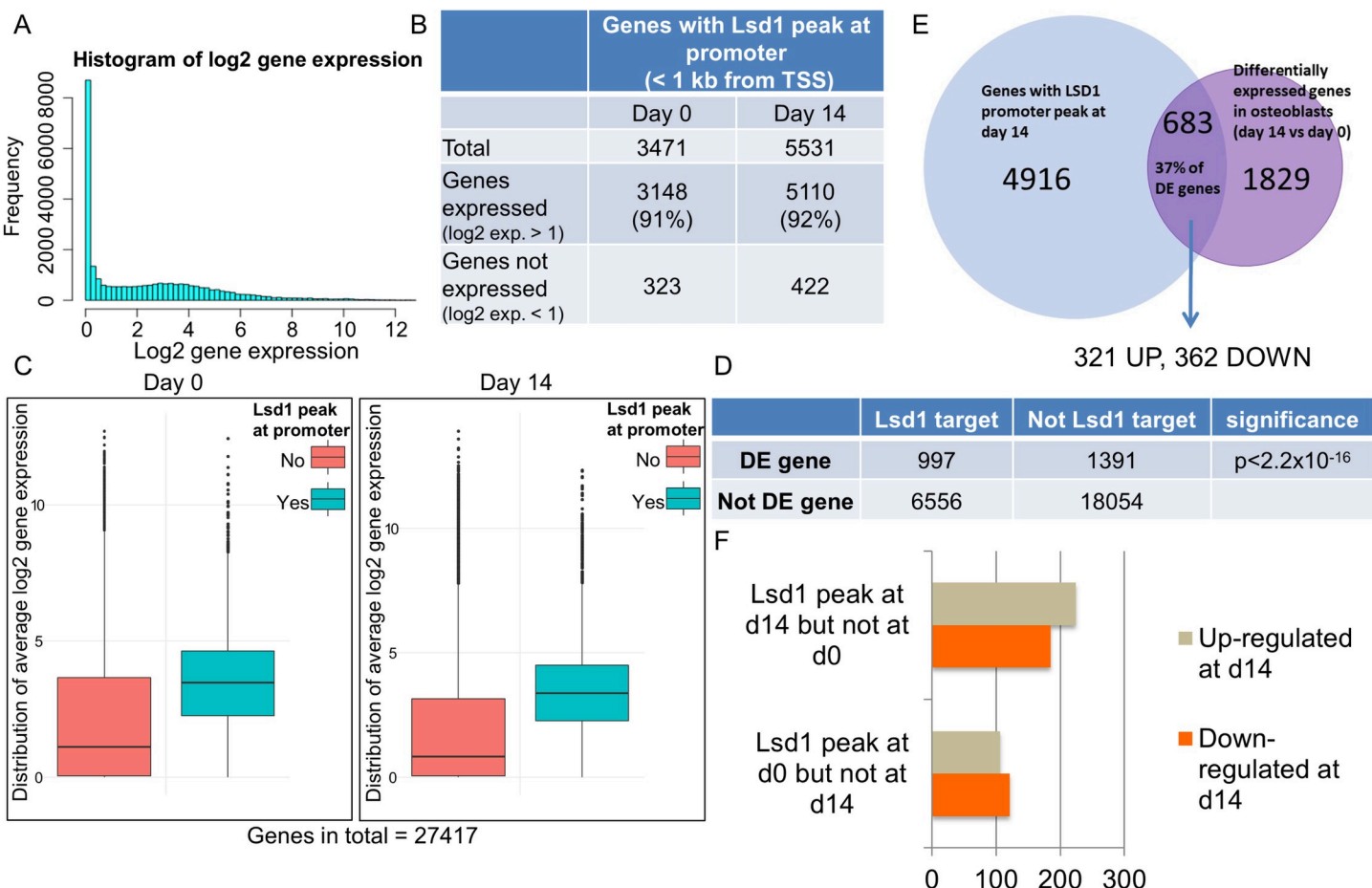

**Fig 9. Lsd1 binding correlates with expression status and differential expression.** (A) Determination of gene expression status of RNA-seq data. Genes with log2 expression level <1 were defined as expressed. (B) Correlation between Lsd1 binding and gene transcription at 0 and 14 d of differentiation. (C) The average expression level of genes with (blue) or without (red) Lsd1 promoter occupancy at 0D and 14D timepoints. (D) Lsd1 binding was significantly enriched on the promoters of DE genes during differentiation. (E) DE genes with Lsd1 occupancy were equally up- and downregulated. (F) Dynamics of Lsd1 binding was not indicative of up- or downregulation of gene transcription.

transcription. However, although Lsd1 binding was abundant on the promoters of expressed genes, Lsd1 was not occupying the promoters of all actively transcribed genes. More specifically, 20% and 33% of the expressed genes at 0 and 14D were occupied by Lsd1, respectively. The latter is consistent with the concept that Lsd1 dynamically associates with its target genes and that Lsd1 may vacate from loci after methylation of H3K4.

We also examined the correlation between dynamics of gene transcription on differentially expressed genes at 0 and 14 d and Lsd1 occupancy. Interestingly, Lsd1 binding was significantly more prominent at the promoters of differentially expressed genes than on the genes with constant expression level between the time-points (Fig 9D). However, these genes contained equal numbers of up- and downregulated genes (Fig 9E), suggesting that Lsd1 binding is not indicative for the specific direction of gene expression changes. Similarly, when we searched for a correlation between Lsd1 binding dynamics and gene expression changes, no signs of Lsd1 binding to up- or down-regulate gene expression were observed (Fig 9F). Taken together, although Lsd1 binding strongly correlates with transcriptional activity and dynamic gene expression changes in differentiated versus undifferentiated cells, the molecular function of this protein in control of gene expression may critically depend on local chromatin context and interactions with other proteins.

## Discussion

Posttranslational histone modifications are important for controlling local chromatin structure and gene transcription. In the present study, we demonstrate that Lsd1, by regulating histone methylation profile, plays a crucial role in osteoblast differentiation in vitro and in vivo. Lsd1 is ubiquitously expressed and it has been shown to play a very context specific role in different types of cells, ranging from the regulation of stem cell properties during embryonic development to growth promoting role in several types of malignancies [43, 81, 82] In vivo, global Lsd1 knockout leads to early embryonic lethality [23, 54]. We found that Lsd1 inhibition by either shRNA mediated knockdown or by pharmacological inhibitors suppressed osteoblast differentiation and function in vitro. Furthermore, inhibition of Lsd1 activity in vivo resulted in decreased osteoblast number and activity, thus causing osteopenia. Conditional deletion of Lsd1 in mesenchymal cells led to osteopenia and disruption of growth plate organization. The decreased number of chondrocytes in the growth plate impaired the typical RANKL expression from the growth plate. The collagen composition analysis with Picrosirius red staining showed, that endochondral bone formation from the growth plate increases deposition of mature Collagen I instead of the transient deposition of Collagen III. This finding may explain differences of bone volume observed by µCT and histomorphometric analyses. We also characterized Lsd1 occupancy in osteoblast genome by ChIP-Seq and found a novel functional interaction of Lsd1 with the master regulator of bone formation, Runx2, further highlighting the important role of Lsd1 in osteogenic differentiation.

Beyond its activity as a histone demethylase, LSD1 binds to DNA by forming a complex with the 'Corepressor of the RE1-Silencing Transcription Factor' (CoREST/RCOR1/KIAA0071), which then interacts with DNA and bridges LSD1 with its substrates [75–77]. In addition to LSD1, all RCORs (RCOR1, RCOR2 and RCOR3) interact with histone deacetylases 1 and 2 (HDAC1 and HDAC2) forming a LSD1-RCOR-HDAC1/2 complex with a unique capacity to both demethylate and deacetylate its targets [83, 84].

LSD1 has been previously shown to bind to both enhancer [85] and promoter areas [47, 86] prior to complex activation. We found that Lsd1 binding was especially enriched at gene promoters with Runx2 consensus motifs many of which had previously been shown to bind Runx2 in ChIP-Seq. Hence, the Lsd1-Runx2 functional interaction may at least in part account for how

Lsd1 is recruited to osteogenic genes during osteoblast differentiation. Previously, genome-wide binding of Lsd1 has been studied by ChIP-seq in 3T3-L1 pre-adipocytes [53], C3H10T1/2 embryonic mesenchymal cells [53], and in brown adipose tissue [87]. In 3T3-L1 cells Lsd1 occupancy at gene promoters increased significantly when the cells reached the mature adipocyte stage, which is in accordance with the high promoter occupancy we observed in osteoblasts. However, we also found that Runx2 transcription factor motifs were the most significantly enriched in osteoblasts, whereas in 3T3-L1 adipocytes Lsd1 occupancy was shown to correlate with Nrf1 consensus motifs [53]. Furthermore, Lsd1 was reported to physically interact with Nrf1 in 3T3-L1 cells [53]. In primary brown adipocytes, Lsd1 has been shown to form a complex with Prdm16, leading to suppression of white adipose tissue (WAT) genes and upregulation of mitochondrial function and thermogenesis, favoring brown adipose tissue (BAT) function [87]. Taken together, our results are in agreement with previous studies suggesting that interaction partners are critical in determining the genomic binding and the outcome of Lsd1 activity in a context specific manner.

Runx2 is well known for its interactions with numerous regulatory proteins, including CBFβ, TLE proteins, HES-1, YAP, Smads, C/EBP, Msx2, AP-1 and HDACs [88, 89]. In addition, proteomic studies have shown that Runx2 may interact with many other protein complexes through indirect interactions [90]. Interestingly, Lsd1 has been previously shown to interact with Runx1 in differentiated erythroid cells, but the interaction was not found in cells at undifferentiated state [91]. Our RNA-seq data showed that both Runx1 and Runx2 mRNAs were expressed in osteoblasts with Runx2 being far more highly expressed than Runx1 as MC3T3-E1 cells matured (S1 Fig). Hence, Lsd1 may switch from Runx1 to Runx2 containing complexes during the progression of osteoblast differentiation, possibly through interactions with other transcription factors or scaffolding proteins.

Lsd1 appears to play a role in several different MSC-related cellular events. Our results suggest that Lsd1 promotes osteoblast differentiation at least in part by modulating chromatin status at Runx2 bound sites. Recently Lsd1 and Runx2 gene transcriptional regulation were also associated in a study of C2C12 cells, where Lsd1 was shown to promote myogenic differentiation by suppressing Runx2 gene expression by demethylating H3K4me1/me2 at an enhancer site 200 kb downstream of Runx2 [92]. In our ChIP-seq data, there were several Lsd1 peaks in the vicinity of the Runx2 locus, but we could not identify this specific enhancer site in our data, perhaps reflecting the high context specificity of Lsd1 action and chromatin binding. Based on a Crispr/Cas9 screen of multipotent C2C12 cells, Munehira and collegues also reported that Kdm3b, Kdm6a and Kdm8 had an effect on osteogenic differentiation, whereas Lsd1 was found to be the only one required for myogenic differentiation. Our study using shRNA mediated silencing or pharmacological inhibition of Lsd1 activity in vitro and in vivo indicate that Lsd1 is required for differentiation of both MC3T3-E1 osteoblasts and primary calvaria osteoprogenitor cells in vitro, as well as osteoblast differentiation and function in vivo.

Recent reports from other groups have concluded that LSD1 inhibition promotes osteogenic differentiation of MSCs. For example, this conclusion has been made based on the finding that MAO-B inhibitor pargyline induced differentiation of human adipose-derived mesenchymal stem/stromal cells [93], and pargyline treatment increased ovariectomy-related and age-dependent bone loss in vivo [94]. Furthermore, limb mesenchyme targeted knockout of Lsd1 by Prrx1-Cre was reported to lead to increased bone mass and Lsd1 depletion in vitro was shown to increase MSC differentiation [95]. Pargyline is much less selective for Lsd1 than the Lsd1 inhibitors we used in our study, which could account for differences in the interpretation of the results stemming from effects from other MAO-B enzymes than Lsd1. It should also be noted that previous studies have examined Lsd1 function in uncommitted MSCs, while we focused on lineage-committed osteoblasts. Hence, the possibility arises that even if Lsd1 may prevent osteogenic lineage-commitment of MSCs it can still have an essential role in

osteoblast maturation when the cell fate has already been determined. Previous experiments published on $Lsd1^{-/-}_{Prrx1}$ mice were conducted on a different Lsd1 conditional knockout construct flanking exon 6 coding for a part of the AOD compared to the construct used in this paper, flanking exons 5 and 6 coding for the FAD binding site and AOD. Original reports of these constructs reported only Lsd1 null phenotypes as no truncated Lsd1 forms were found [51, 54]. Sun and colleagues reported that loss of Lsd1 not only increased bone mass and osteoblast differentiation, but decreased body stature, body weight and delayed cartilage development, though no mention of tibia or femur lengths were made. Therefore, the increased bone volume in μCT analysis in their report could be an outcome of the impaired primary spongiosa ossification and remodeling that was observed in this report.

We found Lsd1 occupied genes to be associated e.g. with cell cycle, chromatin organization and cancer whereas in adipocytes, Lsd1 was found especially in the genes related to adipogenesis, electron transport chain and oxidative phosphorylation [53], indicating the context specificity of Lsd1 action. We observed a strong negative correlation between Lsd1 bound regions and H3K4me1 levels, whereas H3K4me2 levels at these sites were unchanged or increased. This result suggests that Lsd1 may be stably associated with H3K4me2 regions but could quickly vacate these loci when H3K4me1 is demethylated to unmethylated H3K4 in MC3T3-E1 cells. Because Lsd1 knockdown does not reduce global H3K4me3 levels, it is apparent that biological effects of Lsd1 on osteoblast maturation are restricted to specific loci within the genome. Our data together with previous literature of other cell types therefore strongly suggest that interactions of Lsd1 and transcription factors with genomic loci are lineage-specific and chromatin context dependent.

In conclusion, our results support the current view that Lsd1 controls the lineage-commitment of MSCs and apparently favor adipogenic over osteogenic cell fate. However, once cells are pre-committed to the osteoblast lineage (e.g., MC3T3-E1 cells and calvarial osteoblasts), optimal expression of Lsd1 and its recruitment to specific loci is necessary to support osteoblast maturation into its final, functional phenotype. These biological distinctions in Lsd1 activity may be controlled by transcription factors and co-regulators that recruit Lsd1 to certain genomic locations, rather than by changes in Lsd1 expression. We propose that modulation of Lsd1 activity might be beneficial for short term applications in MSC-based bone tissue engineering, but that long-term use in treatment of cancer or osteoporosis would not be advisable from the perspective of the important function of Lsd1 in normal osteoblast differentiation.

## Supporting information

**S1 Fig. LSD1, Runx1 and Runx2 are expressed abundantly during osteoblast differentiation.** Lsd1 expression was stable throughout the differentiation culture of both MC3T3-E1 cells (A) as well as mouse calvarial osteoblasts (B) (n = 3). LSD1 protein expression was abundant and increased during osteoblast differentiation (C) in MC3T3-E1 cells. Both Runx1 and Runx2 were expressed throughout osteoblast differentiation, but Runx2 mRNA expression was clearly higher than Runx1 in the RNA-seq data (n = 2).
(TIFF)

**S2 Fig. Picrosirius red staining analysis showed no changes in collagen maturation under polarized light between groups.** Histological analysis of the Picrosirius red stained tibial sections showed increased collagen I and decreased collagen III staining in the $Lsd1^{-/-}_{Prrx1}$ mice compared to control when normalized to bone area, but the ratio of collagen I and III was unchanged (n = 4 per group). P-values for statistically significant differences are marked * $P<0.05$, ** $P<0.01$.
(TIFF)

**S3 Fig. Kdm7a gene locus H3K4 methylation and LSD1 binding patterns visualized by integrative genomic viewer (IGV).** Kdm7a is an upregulated during differentiation and show Lsd1 binding both at proximal promoter and at upstream enhancer.
(TIFF)

**S4 Fig. Examples of H3K4 methylation patterns at Lsd1 bound promoters and enhancers visualized by integrative genomic viewer (IGV).** Osteogenic locuses for Sp7 (A), Bglap (B) and both Dmp1 and Ibsp (C) show correlation between high Lsd1, low H3K4me1 and high H4K4me2 at 14D.
(TIFF)

**S1 Table. Primer and antibody information.**
(TIFF)

**S2 Table. Lsd1 ChIP-seq peak locations and genes.**
(XLSX)

## Acknowledgments

We are grateful to the members of our laboratories including Roman Thaler and Chris Paradise for stimulation discussions, as well as sharing their expertise and reagents and Merja Lakkisto for her constant efforts in providing reliable RNA and DNA material from the tissue samples. We also thank Asha Nair and Jean-Pierre Kocher, as well as Tamas Ordog for assisting with bioinformatic and epigenomic analyses.

## Author Contributions

**Conceptualization:** Kati Tarkkonen, Amel Dudakovic, R. David Hawkins, Andre J. van Wijnen, Riku Kiviranta.

**Data curation:** Petri Rummukainen, Kati Tarkkonen, Andre J. van Wijnen, Riku Kiviranta.

**Formal analysis:** Petri Rummukainen, Kati Tarkkonen, Amel Dudakovic, Rana Al-Majidi, Vappu Nieminen-Pihala, Cristina Valensisi, R. David Hawkins, Andre J. van Wijnen, Riku Kiviranta.

**Funding acquisition:** Andre J. van Wijnen, Riku Kiviranta.

**Investigation:** Petri Rummukainen, Kati Tarkkonen, Amel Dudakovic, Rana Al-Majidi, Vappu Nieminen-Pihala, Cristina Valensisi, R. David Hawkins.

**Methodology:** Petri Rummukainen, Kati Tarkkonen, Amel Dudakovic.

**Resources:** Petri Rummukainen, Andre J. van Wijnen, Riku Kiviranta.

**Supervision:** Andre J. van Wijnen, Riku Kiviranta.

**Validation:** Petri Rummukainen, Kati Tarkkonen.

**Visualization:** Petri Rummukainen, Kati Tarkkonen, Rana Al-Majidi, Vappu Nieminen-Pihala, Cristina Valensisi, R. David Hawkins.

**Writing – original draft:** Petri Rummukainen, Kati Tarkkonen, Amel Dudakovic, Rana Al-Majidi, Vappu Nieminen-Pihala, Cristina Valensisi, R. David Hawkins, Andre J. van Wijnen, Riku Kiviranta.

**Writing – review & editing:** Petri Rummukainen, Kati Tarkkonen, Amel Dudakovic, Andre J. van Wijnen, Riku Kiviranta.

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
