## [Decision Letter · Decision Letter 0]

7 Jan 2022

PONE-D-21-38069Lysine-Specific Demethylase 1 (LSD1) epigenetically controls osteoblast differentiationPLOS ONE

Dear Dr. Rummukainen,

Thank you for submitting your manuscript to PLOS ONE. After careful consideration, we feel that it has merit but does not fully meet PLOS ONE’s publication criteria as it currently stands. Therefore, we invite you to submit a revised version of the manuscript that addresses the points raised during the review process.

We look forward to receiving your revised manuscript.

Kind regards,

Dengshun Miao

Academic Editor

PLOS ONE

Journal Requirements:

"This study was funded by the Academy of Finland (298625, 268535, and 139165 to RK), as well as foundation grants (Emil Aaltonen Foundation, Sigrid Juselius Foundation and Finnish Cultural Foundation). Support was also provided by the National Institutes of Health (R01-AR09069 to AJvW). "

"RK: 298625, Academy of Finland, https://www.aka.fi/

RK: 268535, Academy of Finland, https://www.aka.fi/

RK: 139165, Academy of Finland, https://www.aka.fi/

AvW: R01-AR09069, National Institutes of Health, https://www.nih.gov/

Reviewers' comments:

Reviewer's Responses to Questions

**Comments to the Author**

1. Is the manuscript technically sound, and do the data support the conclusions?

Reviewer #1: Yes

Reviewer #2: Yes

2. Has the statistical analysis been performed appropriately and rigorously? 

Reviewer #1: Yes

Reviewer #2: Yes

3. Have the authors made all data underlying the findings in their manuscript fully available?

Reviewer #1: Yes

Reviewer #2: Yes

4. Is the manuscript presented in an intelligible fashion and written in standard English?

Reviewer #1: Yes

Reviewer #2: Yes

5. Review Comments to the Author

Reviewer #1: In this study, they examined roles of LSD1 in the osteoblast phenotypes in vitro studies using MC3T3-E1 cells and in vivo mouse study.

They showed that LSD1 inhibition suppresses the expression of osteogenic genes, ALP statin and mineralization in MC3T3-E1 cells. Moreover, they confirmed the roles of LSD1 using Prrx1-Cre LSD1 deleted mice. Moreover, they presented some evidence of the epigenetic regulation of LSD1 in osteoblast differentiation.

They presented enough detailed data supporting their conclusion, which will result in the addition and extension of previously reported studies. The manuscript and figures are generally well prepared.

There are a few minor points, which might be addressed.

1. In some figures, number of samples and significant asterisks are not adequately shown, which should be added carefully.

2. The origin of MC3T3-E1 cells should be written clearly.

3. Re: Line 359, Data should be shown in supplemental figure.

4. BMD data could be added in Figures 3 and 4.

5. Some syntactic errors of English should be carefully corrected. For example, the second sentence of Abstract.

Reviewer #2: Petri Rummukainen et al. investigate the function of lysine-specific demethylase 1 (LSD1) in osteoblast differentiation. Mechanistically, they find Lsd1 occupies Runx2-binding cites at H3K4me2 and H3K4me3 that has an essential role for bone formation. The experiments are decently performed, and the conclusions are solid. I only have a few comments below.

1. Fig 1: Western blot assay will be needed to show the protein levels of LSD1 during osteoblast differentiation.

2. Fig 1 and Fig 2: The bar figures should be correctly labeled if they have statistically significance.

3. Fig 3F: a representative image should be provided, as calcein double-labelling assay is used to be measured the distance between the two dyes of the indicated time.

6. PLOS authors have the option to publish the peer review history of their article (what does this mean?). If published, this will include your full peer review and any attached files.

Reviewer #1: No

Reviewer #2: No

---

## [Author Response · Author response to Decision Letter 0]

17 Feb 2022

PONE-D-21-38069

Lysine-Specific Demethylase 1 (LSD1) epigenetically controls osteoblast differentiation

PLOS ONE

We would like to thank the editor and reviewers for the peer-review of our manuscript as well as excellent comments and suggestions. We have carefully reviewed the editorial as well as reviewers’ comments on our manuscript, performed additional analysis and modified the manuscript accordingly to address these important points raised. Please find our detailed, point to point responses below.

Journal Requirements:

Thank you for this point. We have thoroughly gone through the style requirements and edited the manuscript accordingly.

"This study was funded by the Academy of Finland (298625, 268535, and 139165 to RK), as well as foundation grants (Emil Aaltonen Foundation, Sigrid Juselius Foundation and Finnish Cultural Foundation). Support was also provided by the National Institutes of Health (R01-AR09069 to AJvW). "

"RK: 298625, Academy of Finland, https://www.aka.fi/

RK: 268535, Academy of Finland, https://www.aka.fi/

RK: 139165, Academy of Finland, https://www.aka.fi/

AvW: R01-AR09069, National Institutes of Health, https://www.nih.gov/

Thank you for this valuable correction. We have removed the funding information from the manuscript and would like to edit the Funding Statement to following:

"RK: 298625, Academy of Finland, https://www.aka.fi/

RK: 268535, Academy of Finland, https://www.aka.fi/

RK: 139165, Academy of Finland, https://www.aka.fi/

KT: 275001, Academy of Finland, https://www.aka.fi/

AvW: R01-AR09069, National Institutes of Health, https://www.nih.gov/

We have edited the manuscript accordingly, complementing this manuscript with one new supplemental figure (now S1 Figure) to bring forward this additional data as well as removing one reference to unshown data.

Captions for Supporting Information files have been moved to the end of the manuscript from their previous locations.

Thank you for this important note. Reference list has been reviewed and analyzed by scite.ai tool, which determines whether or not the references listed have received any editorial notices. Of the 95 references in this manuscript, 3 articles referred had been corrected or amended post the original publication:

40. Ye L, Fan Z, Yu B, Chang J, Al Hezaimi K, Zhou X, et al. Histone demethylases KDM4B and KDM6B promotes osteogenic differentiation of human MSCs. Cell Stem Cell. 2012 Jul;11(1):50–61: A duplicated reference photo had been replaced with the correct images, not impacting the interpretation of the results.

43. Whyte WA, Bilodeau S, Orlando DA, Hoke HA, Frampton GM, Foster CT, et al. Enhancer decommissioning by LSD1 during embryonic stem cell differentiation. Nature. 2012 Feb;482(7384):221–5: A duplicated western blot image was corrected to represent the actual images, not altering the analyzed tubulin level. Also black lines were added for extra clarity on lane recognition.

81. Yu Y, Schleich K, Yue B, Ji S, Lohneis P, Kemper K, et al. Targeting the Senescence-Overriding Cooperative Activity of Structurally Unrelated H3K9 Demethylases in Melanoma. Cancer Cell [Internet]. 2018 Feb 12;33(2):322-336.e8: The name of author Mark R. Silvis’s name was spelled incorrectly as Mark S. Silvis. Other references did not flag any editorial concerns in the scite.ai analysis and the corrections in these mentioned articles do not impact the value of their science for this manuscript.

Reviewer #1: In this study, they examined roles of LSD1 in the osteoblast phenotypes in vitro studies using MC3T3-E1 cells and in vivo mouse study.

They showed that LSD1 inhibition suppresses the expression of osteogenic genes, ALP statin and mineralization in MC3T3-E1 cells. Moreover, they confirmed the roles of LSD1 using Prrx1-Cre LSD1 deleted mice. Moreover, they presented some evidence of the epigenetic regulation of LSD1 in osteoblast differentiation.

They presented enough detailed data supporting their conclusion, which will result in the addition and extension of previously reported studies. The manuscript and figures are generally well prepared.

There are a few minor points, which might be addressed.

1. In some figures, number of samples and significant asterisks are not adequately shown, which should be added carefully.

We warmly thank the reviewer for these comments. The number of samples and significances have been added where missing and applicable.

2. The origin of MC3T3-E1 cells should be written clearly.

Thank you for this important point. The origin of MC3T3-E1 cells has been added to material and methods: 

Mouse MC3T3-E1 cells were purchased from ATCC (MC3T3-E1 Subclone 4, CRL-2593) and maintained in �MEM supplemented with 100 U/ml penicillin-streptomycin, 10% fetal calf serum and 2 mM L-glutamine (maintenance medium).

3. Re: Line 359, Data should be shown in supplemental figure.

We thank the reviewer for this comment and this information has been added to the new S1 figure.

4. BMD data could be added in Figures 3 and 4.

Cortical BMD data is available in Figures 3 and 4 labeled “uCT cort BMD”. BMD of trabecular bone, often reported as tissue mineral density or TMD is not reported here as it is not a part of our regular analysis pattern due to the difficulties of defining a ROI in such a multifaceted surface and higher structural variability.

5. Some syntactic errors of English should be carefully corrected. For example, the second sentence of Abstract.

Attention has been directed to syntactic errors in this article. Fixed second sentence of abstract to “Histone methylation is controlled by multiple lysine demethylases and is an important step in controlling local chromatin structure and gene expression.”

Reviewer #2: Petri Rummukainen et al. investigate the function of lysine-specific demethylase 1 (LSD1) in osteoblast differentiation. Mechanistically, they find Lsd1 occupies Runx2-binding cites at H3K4me2 and H3K4me3 that has an essential role for bone formation. The experiments are decently performed, and the conclusions are solid. I only have a few comments below.

1. Fig 1: Western blot assay will be needed to show the protein levels of LSD1 during osteoblast differentiation.

We thank the reviewer for their important comments on this paper. In the previous version of this manuscript the LSD1 expression was shown only on mRNA level using both RNA-seq and qPCR level due to the abundant RNA expression levels as well as difficulties of extracting proteins from differentiated osteoblast cultures where extracellular matrix formation is extensive. Data has now been added to the new S1 figure.

2. Fig 1 and Fig 2: The bar figures should be correctly labeled if they have statistically significance.

We thank the reviewer for this important point. Labels for statistical significance has been added to figures 1 and 2.

3. Fig 3F: a representative image should be provided, as calcein double-labelling assay is used to be measured the distance between the two dyes of the indicated time.

Thank you for this good point. The images chosen to Figure 3F represent accurately the label presence in these samples. In all of our samples the presence of double labeling is confirmed from the cortical bone before analysis. Our current microscope setup does not accurately visualize the different colors of labels where double labels are present, where calcein label is dominantly green but demeclocycline also appears as a milder green label. To accurately measure both single- and double labeling in these samples, we used an alternative filter to visualize only demeclocycline and then combine this information to the calcein label data to allow for an accurate, precise analysis.

---

## [Editor Report · Decision Letter 1]

21 Feb 2022

Lysine-Specific Demethylase 1 (LSD1) epigenetically controls osteoblast differentiation

PONE-D-21-38069R1

Dear Dr. Rummukainen,

We’re pleased to inform you that your manuscript has been judged scientifically suitable for publication and will be formally accepted for publication once it meets all outstanding technical requirements.

Kind regards,

Dengshun Miao

Academic Editor

PLOS ONE
---

## [Editor Report · Acceptance letter]

24 Feb 2022

PONE-D-21-38069R1 

Lysine-Specific Demethylase 1 (LSD1) epigenetically controls osteoblast differentiation 

Dear Dr. Rummukainen:

I'm pleased to inform you that your manuscript has been deemed suitable for publication in PLOS ONE. Congratulations! Your manuscript is now with our production department. 

Kind regards, 

on behalf of

Dr. Dengshun Miao 

Academic Editor

PLOS ONE